# LLM Processes: Numerical Predictive Distributions Conditioned on Natural Language

**James Requeima**[*]
University of Toronto
Vector Institute
requeima@cs.toronto.edu

**John Bronskill**[*]
University of Cambridge
jfb54@cam.ac.uk

**Dami Choi**
University of Toronto
choidami@cs.toronto.edu

**Richard E. Turner**
University of Cambridge
The Alan Turing Institute
ret26@cam.ac.uk

**David Duvenaud**
University of Toronto
Vector Institute
duvenaud@cs.toronto.edu

## Abstract

Machine learning practitioners often face significant challenges in formally integrating their prior knowledge and beliefs into predictive models, limiting the potential for nuanced and context-aware analyses. Moreover, the expertise needed to integrate this prior knowledge into probabilistic modeling typically limits the application of these models to specialists. Our goal is to build a regression model that can process numerical data and make probabilistic predictions at arbitrary locations, guided by natural language text which describes a user's prior knowledge. Large Language Models (LLMs) provide a useful starting point for designing such a tool since they 1) provide an interface where users can incorporate expert insights in natural language and 2) provide an opportunity for leveraging latent problem-relevant knowledge encoded in LLMs that users may not have themselves. We start by exploring strategies for eliciting explicit, coherent numerical predictive distributions from LLMs. We examine these joint predictive distributions, which we call LLM Processes, over arbitrarily-many quantities in settings such as forecasting, multi-dimensional regression, black-box optimization, and image modeling. We investigate the practical details of prompting to elicit coherent predictive distributions, and demonstrate their effectiveness at regression. Finally, we demonstrate the ability to usefully incorporate text into numerical predictions, improving predictive performance and giving quantitative structure that reflects qualitative descriptions. This lets us begin to explore the rich, grounded hypothesis space that LLMs implicitly encode.

## 1 Introduction

Incorporating prior knowledge into predictive models is highly challenging which can restrict the scope for detailed, context-sensitive analysis. In addition, the skill required to incorporate this prior knowledge into probabilistic modelling can restrict the use of these models to experts. In this work, our objective is to develop a probabilistic prediction model that facilitates user interaction through straightforward, natural language. For this purpose, we explore strategies for eliciting explicit, coherent numerical predictive distributions from LLMs.

---

[*]Equal contribution.

38th Conference on Neural Information Processing Systems (NeurIPS 2024).

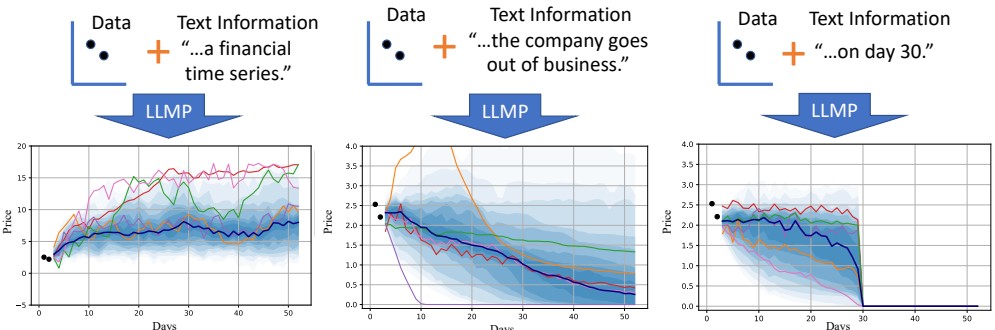

Figure 1: Predictive distributions from an LLMP conditioned on both data and text information. The tenth-percentiles from 50 samples are visualized in faded blue and the median is presented in dark blue with five random samples shown in various colours.

Why go to so much effort to elicit predictions from a slow, expensive, and sometimes inconsistent model like an LLM? We expect their hypothesis class to be both rich, and grounded in exactly the kinds of high-level side information that we currently struggle to communicate to our numerical models. For instance, knowing that prices rarely go below zero, that certain kinds of sensors can saturate at particular values, or that trends almost always eventually level off, are easy to express in natural language, but not straightforward to incorporate into a model without getting lost in difficult-to-specify details about aspects of the domain that aren't well understood. To summarize, we want to develop such a model because it would allow users to 1) provide prior, potentially expert, information to the model about the problem setting in plain-language rather than attempting to capture this information in closed form priors (e.g. Gaussian Process kernels) and 2) it would allow users to access problem-relevant latent knowledge encoded in LLMs that users may not have themselves.

LLMs have recently been shown to be able to condition on the particular task being solved, leveraging contextual information to make better predictions or decisions [1]. They have also been shown to competitively predict time series based only on a text tokenization of numerical data [2]. In this work, we further push in both these directions; 1) using LLMs for numerical prediction tasks going beyond one-dimensional time series forecasting to multi-dimensional regression and density estimation and 2) exploring the ability of these models to condition on both numerical data and rich, unstructured text to improve these predictions. In this paper we make the following contributions:

- **We define LLM Processes (LLMPs) using methods we develop for eliciting numerical predictive distributions from LLMs.**[2] LLMPs go beyond one-dimensional time series forecasting to multi-dimensional regression and density estimation. We propose two approaches for defining this joint predictive distribution over a collection of query points and evaluate their compatibility in principle with the consistency axioms necessary to specify a valid statistical process.

- **We develop effective prompting practices for eliciting joint numerical predictions.** We investigate various methods for conditioning LLMs on numerical data, including prompt formatting, ordering, and scaling. We characterize which schemes perform best on a set of synthetic tasks.

- **We show that LLMPs are competitive and flexible regressors even on messy data.** Through an extensive set of synthetic and real world experiments, including image reconstruction and black-box function optimization, we evaluate the zero-shot regression and forecasting performance of LLMPs. We demonstrate that LLMPs have well-calibrated uncertainty and are competitive with Gaussian Processes (GPs), LLMTime [2], and Optuna [3]. We show that LLMPs use in-context learning to automatically leverage information from related datasets, can easily handle missing datapoints, perform image reconstruction, and output multimodal predictive distributions.

- **Lastly, we demonstrate the ability to usefully incorporate problem-relevant information provided through unstructured text into numerical predictions**, visualized in Figure 1, resulting in quantitative structure that reflects qualitative descriptions. Other additions such as labelling features using text and specifying units allow LLMPs to make use of usually-ignored side information.

---

[2]Source code available at: `https://github.com/requeima/llm_processes`

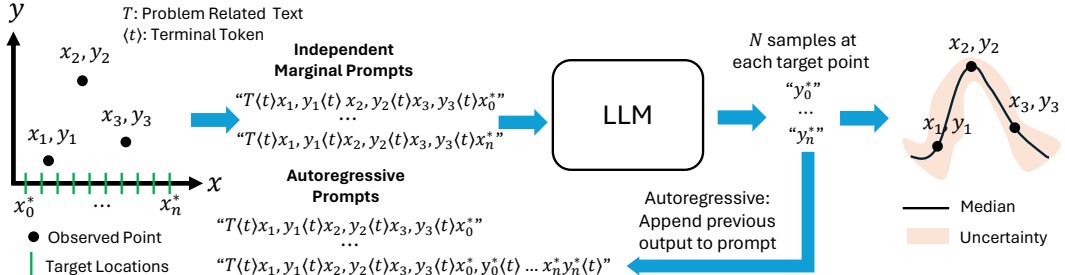

Figure 2: Sampling from an LLM using either independent marginal or autoregressive sampling.

## 2 LLM Processes: Defining a Stochastic Process That Can Condition on Text

Our goal for this section is to use an LLM to elicit joint predictive distributions over arbitrary sized target sets that we can guide and modify using natural language. Formally, given a set of input and output observations $D_{\text{train}} = \{(x_i, y_i)\}_{i=1}^M$ and some text, $T$, we would like to elicit the predictive distribution defined by an LLM at a collection of targets $\{(x_j^*, y_j^*)\}_{j=1}^N$ denoted $p_{\text{LLM}}(y_1^*, \ldots, y_N^* \mid x_1^*, \ldots, x_N^*, D_{\text{train}}, T)$.

Rejection sampling from an LLM allows us to access what we may interpret as the LLM's predictive distribution and gain insights into the model's inductive biases; sampling from the LLM's categorical distribution over text tokens while ignoring non-numerical tokens yields numerical samples from the LLM. The process of sampling from an LLM is depicted in Figure 2 and Algorithm 1. Sample prompts are in Appendix C. Since an accurate sampling-based empirical distribution incurs a high computational cost, next we define an approach to elicit continuous likelihoods from an LLM.

**Continuous Marginal Likelihoods From an LLM.** We approximate a continuous density over our target values by discretizing the space using bins with arbitrarily fine precision, similar to the method used in Gruver et al. [2]. Crucially, this hierarchical approach allows us to compute the probability of a bin with width $10^{-n}$. For example, if $n = 1$ then $\Pr\{y \in [1.0, 1.1)\} = p(1)p(.|1)p(0|1.)$ because '1.0' is a prefix for all $y \in [1.0, 1.1)$. We can convert probability mass to probability density by assuming a uniform distribution within each bin, and dividing the mass by the bin width. A visualization of this construction is in Figures G.2 to G.4.

Unlike [2], we do not rescale the values to remove decimal places. We hypothesize that such scaling removes prior information communicated to the LLM via the scale of the problem. We examine the effect of scaling values in Section 3. We also differ from [2] by including a terminal token after every value in our prompt – for example, given a terminal token $\langle t \rangle$, we represent 12 as $12\langle t \rangle$. Including a terminal token prevents numbers of varying orders of magnitude to share the same prefix – i.e. $p(1)p(2|1)p(\langle t \rangle|12)$ no longer includes the probability of numbers in $[120, 130)$, $[1200, 1300)$, etc.

Note that this approach does not guarantee that $P(12\langle t \rangle)$ yields the mass assigned by the LLM to values in the bin $[12, 13)$ but we empirically observed that our predictive distribution closely matches the sampling distribution to our satisfaction. See Section G.1 for more details and comparison.

**Defining an LLM Process.** Thus far we have established a procedure defining the predictive distribution at a single target location, $p_{\text{LLM}}(y_n^* \mid x_n^*, D_{\text{train}}, T)$. We now outline two methods which we call independent marginal (I-LLMP) and autoregressive (A-LLMP) predictions, for defining the joint predictive distribution over a collection of target points:

$$p_{\text{I-LLMP}}(y_1^*, \ldots, y_N^* \mid x_1^*, \ldots, x_N^*, D_{\text{train}}, T) = \prod_{n=1}^N p_{\text{LLM}}(y_n^*, \mid x_n^*, D_{\text{train}}, T) \tag{1}$$

$$p_{\text{A-LLMP}}(y_1^*, \ldots, y_N^* \mid x_1^*, \ldots, x_N^*, D_{\text{train}}, T) = \prod_{n=1}^N p_{\text{LLM}}(y_n^* \mid y_1^*, \ldots, y_{n-1}^*, x_1^*, \ldots, x_n^*, D_{\text{train}}, T) \tag{2}$$

We note that Equation (1) satisfies the Kolmogorov Extension Theorem [4] therefore defining valid stochastic process (see Appendix A.3). However, it assumes conditional independence given the training set and model weights and the stochastistity represented by the model is via independent marginals. Equation (2) takes inspiration from the autoregressive structure of the LLMs predictive distribution and should yield much richer predictive distributions as we are now able to model

dependencies between output variables. However, this definition is no longer guaranteed to give us a valid stochastic process as the predictive distribution is now target order dependent and will likely fail the Kolmogorov exchangability condition. We investigate both of these questions in Section 3.

**Connection to Neural processes** Neural Processes (NPs) [5] are a class of meta-learning models parametrized by neural networks and trained to learn a map from training (context) sets to predictive distributions, $p_\theta(y_1^*, \ldots, y_N^* \mid x_1^*, \ldots, x_N^*, D_{\text{train}})$. The definitions in Equations 1 and 2 take inspiration from the joint distributions defined by Conditional NPs [5] as independent marginals conditioned on the training/context set and Autoregressive NPs [6] utilizing the chain rule of probability, respectively. Through this lens, LLMPs can be viewed as examples of NPs. However, NPs are directly trained to output this predictive distribution where as LLMPs are repurposing pretrained LLMs.

**Multi-dimensional Density Estimation and Handling Missing Data.** We highlight that, through the flexibility of the LLM prompt, we do not have to draw a distinction between which variables, or variable dimensions are to be modelled or conditioned and can easily handle missing values. Suppose we have a collection of variables $\{x_1, \ldots, x_n\}$ and $\{y_1, \ldots, y_m\}$ (or more), some subset of which we would like to regress on (including $x$ and $y$-values) and the remainder we wish to condition on. To do so using an LLMP, we simply construct the training prompt such that the variables we would like to regress on occur at the end of the prompt and are blank (generated) when sampling from the LLMP. If any values are missing they can simply be removed from the prompt.

## 3  LLMP Configuration

**Experiment Details.** In all of the experiments in Sections 3 to 5, we use six different open source LLMs: Mixtral 8×7B, Mixtral-8×7B-Instruct [7], Llama-2 7B, Llama-2 70B [8], Llama-3 8B, and Llama-3 70B [9]. Note that we never modify the LLM parameters via training or fine-tuning, we use only prompting. Our primary metrics are negative log probabilities (NLL) of the model evaluated at the true function values $f(x^*)$ averaged over the target locations and Mean Absolute Error (MAE) between the predictive median and the true function value. Unless otherwise stated, we use 50 samples from the LLM at each target location $x^*$ and compute the median and the 95% confidence interval of the sample distribution. Details of the datasets are given in Appendix D. Since the LLMs used in our experiments have undisclosed training sets, we address the steps taken to mitigate the issue of data-leakage in Appendix E. Additional implementation details and processing times are in Appendix F.

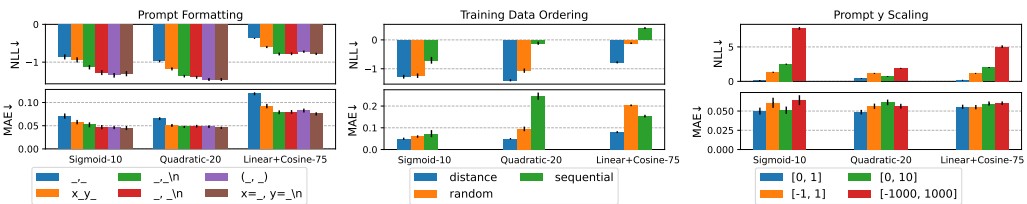

Figure 3: NLL and MAE for various prompt formats ordered from the most to least token efficient (*left*), training data orderings (*middle*), and prompt $y$-scaling (*right*) using the Mixtral-8×7B LLM. The height of each bar is the mean of 10 random seeds that determine the training point locations. The vertical black lines indicate the standard error. In the Prompt Formatting legend (*left*), the two '_' characters indicate the positions of the $x$ and $y$ values and \n represents a new line terminal token.

**Prompt Engineering.** We perform a set of experiments for determining the best LLMP prompt configuration. We use the Sigmoid, Quadratic, and Linear+Cosine functions with 10, 20 and 75 training points, respectively (see Appendix D.1) with I-LLMP using the Mixtral-8×7B LLM.

- *Prompt Formatting* Two separators are required to achieve the best performance. One to separate the $x$ and $y$ values within a pair and another to separate the $x, y$ pairs. Figure 3 (*left*) demonstrates that _,_\n is the best option in terms of performance and token efficiency.
- *Prompt Ordering* Figure 3 (*middle*) shows that ordering the training points by distance to the current target point is best, outperforming both random and sequential ordering. We posit that ordering by distance provides a hint to the LLM to weigh the contribution of closer training points to the current target point to a greater degree.

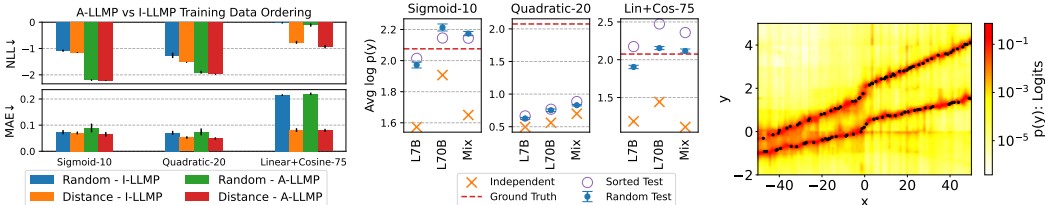

Figure 4: Autoregressive Experiments. *Left:* NLL and MAE for A-LLMP and I-LLMP using different prompt orderings using the Mixtral-8x7B LLM. The height of each bar is the mean of 3 random seeds that determine the training point locations. The black lines indicate the standard error. *Center:* Log-likelihood results of using various test set orderings with Llama-2-7B, Llama-2-70B and Mixtral-8x7B A-LLMP. The orange X indicates I-LLMP, the purple circles used distance ordered test points, and the blue whiskers are the mean and standard error of 10 randomly sampled test orderings. The red dashed line shows the log-likelihood of the test set under the generative process. *Right:* Heatmap visualization of the Llama-3-70B A-LLMP predictive distribution conditioned on data from a bimodal generative process. Black dots are training points.

- *Prompt y-Scaling* Figure 3 (*right*) shows that performance degrades as the range of the $y$ components of the training points increases and when incorporating negative values. This is due to the fact that when the range is wider, the LLM must accurately generate more numerical digits and potentially a negative sign when predicting $f(x^*)$.
- *top-p and Temperature* Figure G.9 shows that performance is surprisingly insensitive to varying the LLM nucleus sampling parameter top-$p$ [10] and LLM softmax temperature.

**Autoregressive vs Independent Marginal Predictions.** Here we examine two questions: first, does the autoregressive defininiton of the joint predictive likelihood (A-LLMP) in Equation (2) improve performance versus the independent marginal definition of Equation (1) (I-LLMP). Second, "how close" is A-LLMP to a stochastic process in terms of performance variability across query orderings.

We first look at log-likelihoods and MAE for A-LLMP and I-LLMP using the random and distance training point orderings discussed earlier. Results can be seen in Figure 4 (*left*). Similar to our findings earlier, ordering the training values according to distance to target has a large effect, improving performance for both I-LLMP and A-LLMP. Unsurprisingly, the richer joint distribution given by A-LLMP gives us better predictive performance.

We next examine the variability in performance of A-LLMP when different autoregressive target orderings are used to get a sense of how far our method is from a stochastic process (which would be permutation invariant in the target points). The results of using ten sets of randomly ordered target points compared to I-LLMP and the ground truth log-likelihood of the test sample under the generative distribution are presented in Figure 4 (*center*). Note that the training data is distance sorted in all cases. We also present the result when ordering target points according to distance to the closest training point, from smallest to largest. We make three key observations: first, log-likelihood performance of all A-LLMP orderings is better than I-LLMP. Second, the variance of random orderings is small on the scale of the log-likelihood of the generative model. And third, distance ordering the targets gives better or at least competitive performance with a random ordering. These results present practitioners a choice: do you care more about using a valid statistical process or obtaining good predictive performance? If it is the latter, you would be better served using A-LLMP.

## 4 Evaluating LLMP Performance on Numerical Data

In this section, we evaluate the performance of LLMPs on purely numerical data in a wide variety of settings. Additional details and results for experiments in this section can be found in Appendix H.

**1D Synthetic Data Experiments.** To show that LLMPs are a viable regression model with well-calibrated uncertainties, we benchmark in Table 1 our A-LLMP method against a GP on the Function Dataset (Appendix D.1). The GP uses an RBF kernel with optimized length scale and noise. The Mixtral-8×7B A-LLMP achieves the lowest negative log-likelihoods averaged over 7 function sizes and 3 seeds on 10 out of 12 of the functions and equal or better MAE on 8 of the functions. Visualizations of the predictive distributions and plots of MAE and A-LLMP are shown in Appendix H.1.

Table 1: Mean and standard error of MAE and NLL averaged over over the seven training set sizes and 3 seeds of each function for Mixtral-8×7B A-LLMP and a GP with an RBF kernel.

| | Metric | Beat | Exp | Gau Wave | Linear | Lin + Cos | Lin x Sine | Log | Quadratic | Sigmoid | Sinc | Sine | X x Sine |
|---|---|---|---|---|---|---|---|---|---|---|---|---|---|
| GP | MAE↓ | 0.33±0.01 | 0.32±0.12 | **0.20±0.02** | 0.11±0.04 | **0.16±0.02** | 0.12±0.03 | 0.09±0.03 | **0.07±0.01** | 0.37±0.05 | 0.08±0.02 | **0.22±0.02** | 12.79±1.07 |
| | NLL↓ | 0.97±0.23 | -1.03±0.31 | **-0.11±0.21** | -1.45±0.22 | **-0.64±0.18** | -1.38±0.22 | -1.57±0.19 | -0.40±0.29 | 0.03±0.21 | -1.44±0.20 | 0.23±0.32 | 12.64±1.42 |
| LLMP | MAE ↓ | **0.31±0.01** | **0.08±0.01** | 0.24±0.01 | **0.05±0.00** | 0.19±0.01 | **0.05±0.00** | **0.04±0.00** | **0.07±0.01** | 0.51±0.04 | 0.08±0.02 | 0.27±0.02 | **12.45±1.37** |
| | NLL↓ | **-0.78±0.03** | **-1.56±0.04** | -0.08±0.08 | **-2.38±0.08** | -0.15±0.10 | **-1.90±0.02** | **-2.20±0.02** | **-1.35±0.03** | **-0.80±0.04** | **-1.96±0.03** | 0.14±0.11 | 3.30±0.23 |

To verify that LLMPs are able to produce non-Gaussian, multimodal predictive distributions we sampled training data from synthetic, multimodal generative distribution (experimental details in Appendix H.2). The Llama-3-70B LLMP predictive distribution is visualized in Figure 4 (*right*).

**Comparison to LLMTime.** Figure 5 demonstrates that A-LLMP yields superior results in terms of MAE and NLL when compared to LLMTime using Llama-2-7B on a forecasting task using the weather dataset (described in Appendix D.2). Additional plots with missing training data are in Appendix H.3. We posit that A-LLMP betters LLMTime due to the fact that 1) A-LLMP naturally handles irregularly spaced $x$ and $y$ data whereas LLMTime uses only regularly spaced $y$ information requiring imputation with NaN values where data is missing; and 2) A-LLMP performs no scaling on $y$ values in contrast to LLMTime that scales data to eliminate the use of decimals and normalize the range of the data and as a result removes information that the LLM can potentially leverage.

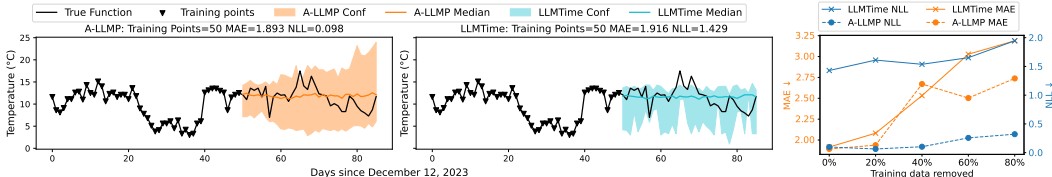

Figure 5: Comparison of A-LLMP and LLMTime on the weather dataset. *Left:* Plot using all 50 training points. *Right:* Plot of MAE and NLL versus the amount of training data removed. A-LLMP has lower MAE and NLL and the margin over LLMTime increases as more training data is removed.

**Comparison to From Words to Numbers.** We compare our I-LLMP method to the approach in [11] on their Original #1 dataset. The experimental set-up is as follows: There are 100 trials with each trial consisting of 50 training points and a single target point. The training and target points for each trial are randomly generated using the function described in [11]. We use the code from their paper to generate the data and evaluate their approach and compare it to ours using identical numerical data. We use the Llama-2-7B LLM for both methods to ensure a fair comparison. I-LLMP achieved lower MAE on 78 of the 100 trials when compared to their method. When the errors are averaged over the 100 trials, the I-LLMP average error was 0.836 and theirs was 3.137. These results indicate that our LLMP approach is clearly superior. This is due to the facts that (i) we sort the training points according to distance to the current target point when creating the prompt whereas they do not, and (ii) we form a distributional estimate for the predicted point and then take the median sample value as the best estimate, whereas they generate a single point estimate.

In the next three experiments we showcase the ability of LLMPs to handle multi-dimensional data.

**Image Reconstruction** As a 2-dimensional input experiment, Figure 6 shows reconstruction results from images drawn from the Fashion-MNIST dataset [12]. We convert pixel data into prompt data points by forming a series of (row, column, pixel value) tuples. Additional results and details are in Appendix H.4. Using 20% train pixels, the basic form is captured and at 50%, the reconstruction is accurate despite the sharp pixel intensity transitions.

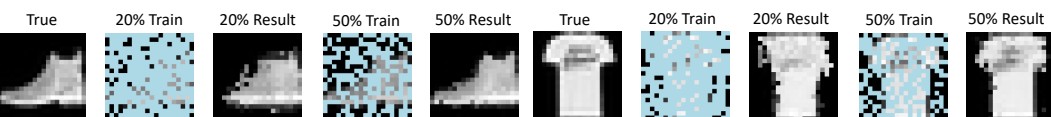

Figure 6: Fashion-MNIST Mixtral image reconstruction results. The blue pixels indicate unobserved.

**Black-Box Function Optimization** Black-box optimization involves minimizing or maximizing a function where there is only access to the output of a function for a specified input. We benchmark the ability of LLMPs to perform maximization on six commonly used multi-dimensional functions. We compare our results using Llama-2-7B to Optuna [3], a commercial hyperparameter optimization framework. Results and implementation details are in Appendix H.5. In all cases, LLMPs obtain as good or better approximation to the true maximum value in a fewer number of trials.

**Simultaneous Temperature, Rainfall, and Wind Speed Regression** To examine how well an LLMP can model multi-dimensional outputs, we compare LLMP regression to a multi-output GP on the weather dataset described in Appendix D.2. Figure 7 shows the results for the Llama-3-8B LLM (*top*) and a 3 output RBF kernel GP with trained hyperparameters (*bottom*). The LLM is similar to and in most cases better than the GP in terms of MAE and NLL.

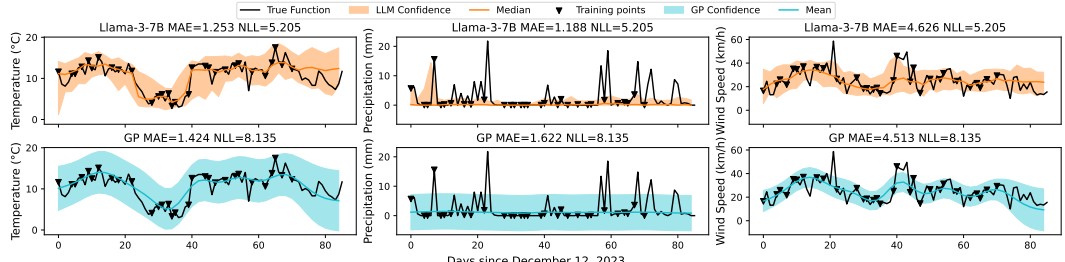

Figure 7: Results for simultaneously predicting temperature, precipitation, and wind speed using the Llama-3-7B LLM (*top*) and a 3 output RBF kernel GP with trained hyperparameters (*bottom*).

**In-context Learning Using Related Data Examples.** In this experiment, we investigate LLMPs' ability to learn from similar examples in-context to predict average monthly precipitation across 13 Canadian locations [13], one from each province and territory. For each location, we use the Mixtral-8×7B A-LLMP to forecast 32 months of average precipitation values given the previous four month observations taken from a random historical three-year period between 1913-2017 (conditional on data availability). It is then provided with 1-12 examples of random three year periods of historical values from the same location in-context. Results shown in Figure 8 and experimental details in Appendix H.6. Conditioning the LLMP on historical examples improves performance saturating after 4 years, and degrading slightly thereafter. Generally, the LLMP is able to use the examples to pick up on seasonal trends from history. We note that some locations do not have obvious or strong seasonal patterns but examples still help performance in these cases (see Appendix H.6).

## 5 Conditioning LLMPs on Textual Information

One of the most exciting directions of LLMPs is the potential to incorporate prior information about problems via text. Now that we can examine functional predictive distributions of LLMs, we can begin to explore their rich prior over functions by conditioning on both text and numerical data. In this section we present two experiments with details and additional experiments presented in Appendix I.

**Scenario-conditional Predictions.** In this experiment, we examine the influence of text providing information about various synthetic problem settings on the predictive distribution of an LLMPs. In

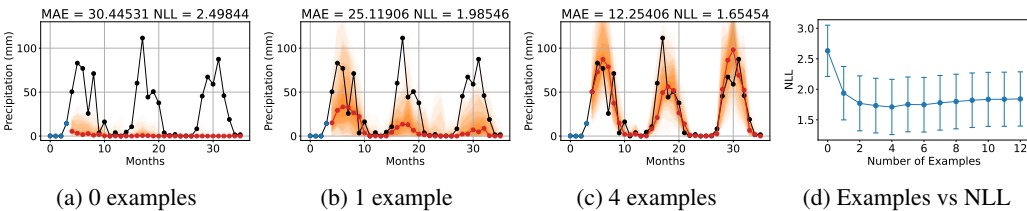

(a) 0 examples      (b) 1 example      (c) 4 examples      (d) Examples vs NLL

Figure 8: (*Left three plots*) Visualizations of the predictions given by the Mixtral-8×7B LLMP for Ranfurly, Alberta. Blue and black circles are training and test points, respectively. Red circles are median predictions and shaded areas indicate tenth-percentiles over 30 samples. (*Right*) NLL vs number of examples. Error bars show standard error over 13 locations.

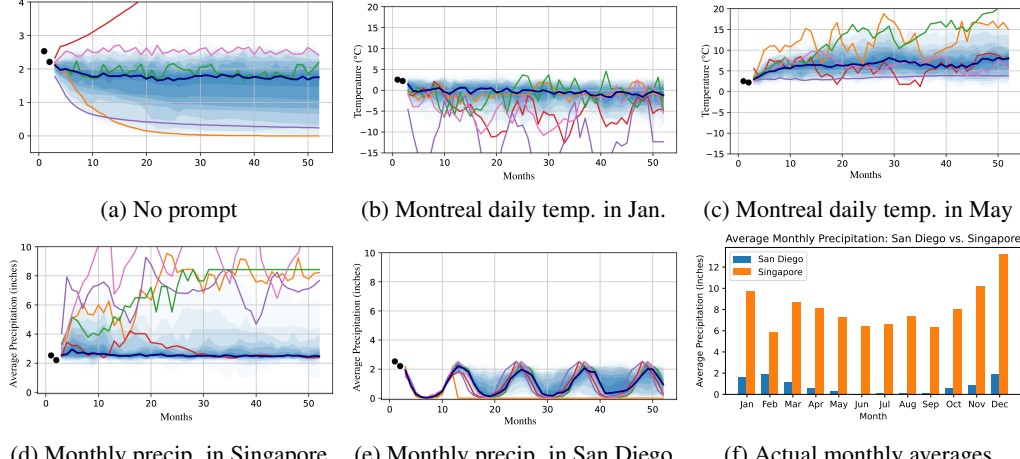

| (a) No prompt | (b) Montreal daily temp. in Jan. | (c) Montreal daily temp. in May |
| (d) Monthly precip. in Singapore | (e) Monthly precip. in San Diego | (f) Actual monthly averages |

Figure 9: a)-e) predictive distributions from an A-LLMP using Llama-3-70B under various scenario prompts. Black points are two training points given to the LLM process, the same values for each scenario. The tenth-percentiles from 50 samples are visualized in faded blue and the median is presented in dark blue with five random samples shown in various colours. Figure f) shows the actual average monthly rainfall for Singapore from 1991-2020 [14] and San Diego from 2000-2024 [15].

all of the following examples, we provide the same two synthetic training points to the LLMP but change the prompting text that comes before the training data. We then use A-LLMP with Llama-3-70B to forecast trajectories 50 steps ahead. We begin by examining the predictive distribution with no prompt (Figure 9a). We prompt the LLMP to generate daily temperature measurements in degrees Celsius from Montreal in January (Figure 9b) and May (Figure 9c), and monthly precipitation values from San Diego, CA (Figure 9d) and Singapore (Figure 9e). Figure 1 Shows the results of prompting the LLMP to generate (*left*) a stock price financial time series (*centre*) for a company that eventually goes out of business and (*right*) for a company whose price goes to zero on day 30.

Indeed, the LLMP modifies the predictive distribution accordingly relative to the no prompt predictions. We highlight the following observations: first, for prompts b) and c), the model moves about half of its predictive mass below zero for temperatures beginning in January and above zero for the May temperatures. Second, the LLMP is able to recall actual historical trends for average monthly precipitation for Singapore and San Diego to condition on prompts d) and e). Despite getting the trend correct, we note that the median prediction in d) seems to be biased toward the training values and not reflective of the actual monthly median.

Last, for stock price simulations, the model places all of its density on positive numbers since it is modelling prices. It is able to produce realistic trajectories and decreases them in expectation when prompted that the company goes out of business. The model is able to condition on the fact that the price goes to zero on day 30 which correctly interprets the meaning of the $x$-values as days starting from 0, that the $y$-axis is the price and the phrase "price goes to zero" corresponds to a $y$-value of 0.

**Labelling Features Using Text.** In the following example, we examine the performance of a Mixtral-8x7B Instruct I-LLMP on predicting American housing prices. The dataset [16] contains 39980 housing prices and various variables around housing and demographics for the top 50 American cities by population. Note that this dataset was generated on 12/09/2023, however it contains data from the 2020 US Census and the 2022 American Community Survey (ACS) so we cannot guarantee that models did not see data within this dataset during training.

For each prediction task, we show the I-LLMP 10 randomly selected training examples from the dataset and predict on 20 randomly selected test examples. In the prompt, before the numerical value (price) we provide a string which encodes the datapoint index/features that the model can use. For our first experiment we examine the behaviour of the LLMP when more features are added to the prompt. We experiment with five ways of indexing the training and test points; For case (1), we provide latitude and longitude of the house as numerical values (eg. 32.74831, -97.21828) converted to strings similar to our method in previous experiments. For the remaining 4 cases, we provide additional labeled features, adding more features for each case with the prompt for case (5) containing

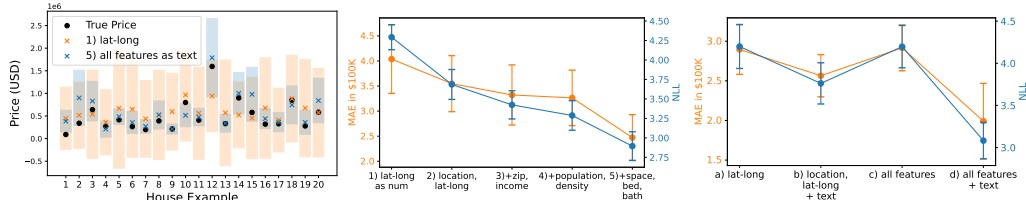

Figure 10: Results of a Mixtral-8x7B Instruct I-LLMP predicting US housing prices. *Left:* Predictions for 10 randomly selected houses using index style 1) and 5). Xs are mean predictions using 30 samples from the LLMP and error bars indicate 2 standard deviations. *Centre and right:* Average MAE and NLL performance of the LLMP over 10 experiments with error bars representing the standard error for experiments from Section 5.

all labelled features, illustrated with the following example: (2) Location: Fort Worth, Texas, Latitude: 32.74831, Longitude: -97.21828, (3) Zip Code: 76112, Median Household Income: 71452.0, (4) Zip Code Population: 42404 people, Zip Code Density: 1445.0 people per square mile, (5) Living Space: 1620 square feet, Number of Bedrooms: 3, Number of Bathrooms: 2.

This procedure is repeated 10 times to compute statistics. Results are presented in Figure 10 (*left, centre*). Note that the LLMP is able to take advantage of the additional features provided to improve predictive performance. To see examine the effect of adding text labels to the features, we ran another set of experiments on 10 new random datasets providing the LLMP with either labeled or unlabelled numerical features. The following are example feature strings: (i) "30.45738, -97.75516" (ii) "Location: Austin, Texas, Latitude: 30.45738, Longitude: -97.75516" (iii) "30.45738, -97.75516, 78729, 107830.0, 30907, 1216.1, 1349, 3" (iv) "Location: Austin, Texas, Latitude: 30.45738, Longitude: -97.75516, Zip Code: 78729, Median Household Income: 107830.0, Zip Code Population: 30907 people, Zip Code Density: 1216.1 people per square mile, Living Space: 1349 square feet, Number of Bedrooms: 3, Number of Bathrooms: 2". Results of this experiment are presented in Figure 10 (*right*). Note that the LLMP is not able to use the raw feature values to improve performance from only 10 training examples, but is able to do so with labelled features suggesting that LLM is able to utilize the latent relationship between the feature and the price once the feature is identified. We found that the Mixtral-8×7B Instruct model had the best performance on this task and was able to utilize text information better (results for other models in Appendix I.2).

# 6    Related Work

In this section, we discuss work related to eliciting distributions from LLMs including forecasting, regression, in-context learning, and nearal processes among others.

**LLM Forecasting** The most closely related work to ours is LLMTime [2]. LLMTime is capable of zero-shot extrapolation of one-dimensional time series data at a level comparable to trained purpose-built approaches. In addition, they develop a method for eliciting marginal probability distribution functions from LLM posteriors over functions, which we build on. They also begin to investigate the effect of conditioning on text. In contrast, we focus on (i) interpolation with multi-dimensional inputs and outputs; (ii) eliciting joint distributions over functions, not just marginals; and (iii) exploring the ability of models to condition simultaneously on both numerical data and text. More recently, TimesFM [17], a foundation model for one-dimensional zero-shot times series forecasting was introduced. However, TimesFM does not support interpolation or higher dimensional data and does not consider distributions. PromptCast [18] performs zero-shot time series forecasting by combining numerical data and text in a question answer format. Our approach for combining problem specific text along with numerical data differs in that it handles both interpolation and extrapolation and does not rely on a question-answer format. Hegselmann et al. [19] utilize LLMs to do zero-shot and few-shot classification on tabular data that compares favorably to standard ML approaches.

**LLM Regression** Pesut [20] do some initial investigations into the use of LLMs as regressors on 1D synthetic functions. Our work greatly expands on these early investigations. Vacareanu et al. [11] is concurrent work that shows that LLMs are capable linear and non-linear regressors. However, their work does not condition on any textual information, compute log probabilities, compare to Gaussian Processes, investigate the effect of prompt formatting, or employ auto-regressive sampling.

**In-context learning (ICL) in LLMs** Xie et al. [21] point out that ICL can be seen as being equivalent to Bayesian inference in a latent variable model. More recently, [22] explain in-context learning in LLMs as kernel regression. Garg et al. [23] train transformers to do in-context learning on various function classes including linear (up to 50 dimensions), decision trees, and two-layer ReLU networks. Coda-Forno et al. [24] demonstrate that LLMs are capable of meta-in-context learning and that performance on 1-D linear regression and two-armed bandit tasks improves with multiple examples. TabPFN [25] is a trained transformer that is able to do tabular classification given in-context examples.

**LLM Hyperparameter Optimization** Zhang et al. [26] and Liu et al. [27] use LLMs to perform hyperparameter optimization, showing that LLMs can condition on a mixture of textual data as numerical observations to effectively optimize hyperparameters in machine learning models.

**Eliciting priors from LLMs** Binz and Schulz [28] fine-tune LLMs on data from psychological experiments to achieve accurate representations of human behavior. Choi et al. [1] show how using an LLM to assess the importance of features or the causal relationship between variables that can improve performance on tasks. Lipkin et al. [29] find that LLMs can derive human-like distributions over the interpretations of complex pragmatic utterances.

**Eliciting distributions from humans** Schulz et al. [30] look at compositional inductive biases in function learning, showing humans have compositional structure in their priors on functions. [31] catalogue standard strategies for eliciting distributions from expert humans.

**Neural processes** Neural Processes are a class of meta-learning models trained to learn a map from training (context) sets to predictive distributions, $p_\theta(y_1^*, \ldots, y_N^* \mid x_1^*, \ldots, x_N^*, D_{\text{train}})$. These models are parameterized using a neural network and there have been various proposals for different architectures using attention [32], transformers [33], Gaussian Process output layers [34], and diffusion models [35]. The definitions of the joint distributions in equations 1 and 2 take inspiration from the joint distributions defined by Conditional Neural Processes [5] as independent marginals conditioned on the training/context set and Autoregressive Neural Processes [6] utilizing the chain rule of probability, respectively. Through this lens, LLMPs can be viewed as examples of Neural Processes. LLMPs differ from standard NPs in two main ways: (i) Training objective: Neural Processes are meta-trained using maximum likelihood to optimize $p(y^*|x^*, D_{\text{train}})$ directly. LLMPs have a very indirect training procedure – they are trained to be language models i.e. autoregressive token predictors. One of the contributions of this paper is the demonstration that, despite this, they can perform zero-shot probabilistic regression. (ii) Architecture: NPs have an output layer that parametrizes the predictive distribution over targets directly. Since LLMPs are repurposing language models for regression, we need to define the mapping from distributions over language tokens to distributions over target variables. We note that LLMs themselves can be viewed as AR-CNPS [6] with a fixed, predefined target ordering.

# 7 Discussion, Limitations, and Societal Impact

Below we discuss our findings, the limitations and societal impact of the work presented. Further discussion on these issues can be found in Appendix J.

**Discussion** We defined LLMPs for eliciting numerical predictive distributions from LLMs and when used as a zero-shot muti-dimensional regression model are competitive with GPs. Excitingly, we demonstrated the ability to condition on text to improve predictions and probe the LLMs' hypothesis space. An interesting extension would be to condition on other modalities in addition to text.

**Limitations** Along with the flexibility of LLMs, LLMPs inherit their drawbacks. Maximum context sizes limit the size of tasks we can apply this method to and the amount of textual information we can condition on. LLMPs are also significantly more computationally expensive compared to Gaussian Processes and standard regression methods. All of experiments were performed on readily available open source LLMs that are smaller and generally less capable compared to proprietary LLMs.

**Societal Impact** Our work has demonstrated a new and useful zero-shot approach for generating probabilistic predictions using plain language to augment numerical data. It has the potential to allow practitioners from fields such as medical research and climate modelling to more easily access probabilistic modelling and machine learning. Like all machine learning technology, there is potential for abuse, and possible consequences from incorrect predictions made with LLMPs. Also, we do not know the biases in the underlying LLMs used and what effect they may have on LLMPs output.

## Acknowledgments and Disclosure of Funding

James Requeima and David Duvenaud acknowledge funding from the Data Sciences Institute at the University of Toronto and the Vector Institute. Dami Choi was supported by the Open Phil AI Fellowship. John Bronskill is supported by EPSRC grant EP/T005386/1. Richard E. Turner is supported by Google, Amazon, ARM, Improbable, EPSRC grant EP/T005386/1, and the EPSRC Probabilistic AI Hub (ProbAI, EP/Y028783/1).

We thank Anna Vaughan for help with the weather datasets and discussions. We also thank Will Tebbutt, Matthew Ashman, Stratis Markou, and Aristeidis Panos for helpful comments and suggestions.

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

# A  LLM Processes: Defining a Stochastic Process That Can Condition on Text

In this section we elaborate on the explanations and definitions in Section 2. Our goal is to use an LLM to elicit joint predictive distribution over arbitrary sized target sets that we can guide and modify using plain language. Formally, given a set of observations $D_{\text{train}} = \{(x_i, y_i)\}_{i=1}^{M}$ and some text, $T$, we would like to elicit the predictive distribution defined by an LLM at a collection of targets $\{(x_j^*, y_j^*)\}_{j=1}^{N}$ denoted $p_{\text{LLM}}(y_1^*, \ldots, y_N^* \mid x_1^*, \ldots, x_N^*, D_{\text{train}}, T)$. To achieve the goal, we can can keep in mind two interpretations of what we mean by a predictive distribution defined by an LLM. First, we can interpret the LLM as maintaining having a predictive distribution over numerical values, which we can probe by sampling from the LLM. This interpretation is beneficial if we believe that the LLM has learned useful prior information that we would like to access via its beliefs about these numerical values and for our goal of guiding the predictive distribution using text. The other interpretation is more empirical: we simply use the LLM as a tool to define a valid predictive distribution and evaluate how well this definition performs on test cases. Our approach is a combination of the two philosophies – we will propose a method defining a predictive distribution that is valid and performs well on test cases, but closely matches what we think of as the LLM's underlying distribution.

## A.1  Continuous Marginal Likelihoods From an LLM

As discussed in Section 2, we use a method similar to the one proposed by Gruver et al. [2]; we approximate the continuous density by discretizing the space using bins with arbitrarily fine precision. Let's assume a fixed number of decimal places $n$, and that LLMs generate one digit at a time[3]. The key idea is that each new digit can be viewed as being generated from a categorical distribution with the probabilities $p$ given by a softmax over numerical tokens. Crucially, this hierarchical approach allows us to compute the probability of a bin with width $10^{-n}$. For example, if $n = 1$ then $\Pr\{y \in [1.0, 1.1)\} = p(1)p(.|1)p(0|1.)$ because '1.0' is a prefix for all $y \in [1.0, 1.1)$ . We can convert probability mass to probability density by assuming a uniform distribution within each bin, and dividing the mass by the bin width. A visualization of this construction can be viewed in Appendix G.1.

The method in [2] has two main shortcomings for our purposes: first, the authors propose to scale all $y \in D_{\text{train}}$ to eliminate decimals from their numerical representation. For example, for a precision of 2 decimal places, the numbers 0.123, 1.23, 12.3, and 123.0 will be transformed to 12, 123, 1230, and 12300 respectively. Scaling removes prior information communicated to the LLM via the scale of the problem. For example, it is likely that the LLM has encountered financial data with decimal places. Potentially, it also makes it more difficult to communicate prior information about the problem to the LLM via text.

Second, probabilities of all sequences of integers given by an LLM contain the mass of all values that also start with that sequence. We can think of this as the problem of not knowing when the LLM intends to terminate a value. For example, if $y = 12$, $\Pr\{y \in [12, 13)\} \neq p(1)p(2|1)$ since $p(1)p(2|1)$ includes the probability of all numbers with '12' as a prefix – this includes [12, 13) but also [120, 130), [1200, 1300) and so on.

## A.2  The LLM Process Method

We follow Gruver et al. [2] and discretize the continuous space with bins of width $10^{-n}$, computing the probabilities for each bin using the hierarchical softmax approach. However, different from their approach we 1) keep values at their original scale, and 2) include a terminal token after every value – for example, given a terminal token $\langle t \rangle$, we represent 12 as $12\langle t \rangle$ and 120 as $120\langle t \rangle$. Including a terminal token prevents numbers of varying orders of magnitude from sharing the same prefix – i.e. $p(1)p(2|1)p(\langle t \rangle|12)$ no longer includes the probability of numbers in [120, 130), [1200, 1300), and so on. After we compute the mass of a bin via hierarchical softmax, we divide the mass by the bin width $10^{-n}$ to get an estimate of the density value. This procedure defines a valid predictive distribution over y-values, and we call this elicitation method 'logit-based' since we derive probabilities from the logits directly instead of sampling. Pseudocode can be found in Algorithm 2.

---

[3]The models we evaluate are trained with tokenization schemes that tokenize each digit in a number separately. Gruver et al. [2] include a space between each digit for tokenizers that do not tokenize each digit separately.

It must be noted that this approach does not guarantee that $P(12\langle t \rangle)$ yields the mass assigned by the LLM to values in the bin $[12, 13]$. However, we note that our method defines a valid predictive distribution and we empirically observed that our predictive distribution closely matches the sampling distribution to our satisfaction (see Appendix G.1).

## A.3 Defining an LLM Process

So far we have established a procedure for defining the predictive distribution at a single target location, $p_{\text{LLM}}(y_n^* \mid x_n^*, D_{\text{train}}, T)$. We now discuss how to define the joint predictive distribution over a collection target points. In particular, we would like to define a stochastic process via its finite-dimensional marginal distributions $\rho_{x_1,\ldots,x_N}$ defined over locations $x_1, \ldots, x_N$. The Kolmogorov Extension Theorem [4] states that such a collection defines a stochastic process if it satisfies

1. *Exchangeability:* Given any permutation $\pi$ of the integers $\{1, \ldots, N\}$

$$\rho_{x_1,\ldots,x_N}(y_1, y_N) = \rho_{x_{\pi(1)},\ldots,x_{\pi(N)}}(y_{\pi(1)}, y_{\pi(N)})$$

2. *Consistency:* if $1 \leq M \leq N$ then

$$\rho_{x_1,\ldots,x_M}(y_1, \ldots, y_M) = \int \rho_{x_{\pi(1)},\ldots,x_{\pi(N)}}(y_{\pi(1)}, y_{\pi(N)}) \, \mathrm{d}y_{M+1} \ldots \mathrm{d}y_N$$

In Equation (1) we define a collection of joint distributions by defining a factorized distribution over target locations $x_1^*, \ldots, x_N^*$:

$$p_{\text{I-LLMP}}(y_1^*, \ldots, y_N^* \mid x_1^*, \ldots, x_N^*, D_{\text{train}}, T) = \prod_{n=1}^{N} p_{\text{LLM}}(y_n^*, \mid x_n^*, D_{\text{train}}, T)$$

where $p_{\text{LLM}}(y_n^*, \mid x_n^*, D_{\text{train}}, T)$ is defined above.

This definition satisfies the Kolmogorov Extension Theorem and so it defines a valid stochastic process. However, it assumes conditional independence given the training set and model weights and, conditional on these variables, the stochastisity represented by the model is via independent marginals. Taking inspiration from the autoregressive structure of the LLMs predictive distribution, we can write the joint distribution according to the product rule:

$$p_{\text{A-LLMP}}(y_1^*, \ldots, y_N^* \mid x_1^*, \ldots, x_N^*, D_{\text{train}}, T) = \prod_{n=1}^{N} p_{\text{LLM}}(y_n^* \mid y_1^*, \ldots, y_{n-1}^*, x_1^*, \ldots, x_n^*, D_{\text{train}}, T)$$

Where, the previous target location is autoregressively added to the conditioning data via the LLM prompt. This should yield much richer predictive distributions as we are now able to model dependencies between output variables. However, this definition is no longer guaranteed to give us a valid stochastic process as the predictive distribution is now target order dependent and most likely will fail the Kolmogorov exchangability condition. We investigate these questions in Section 3.

# B LLM Processes Pseudocode

---

**Algorithm 1** Pseudocode for sampling numbers from an LLM

---

$N \leftarrow$ Number of desired samples
samples $\leftarrow$ [ ]
**while** len(samples) $< N$ **do**
    out $\leftarrow$ model.generate(prompt)
    **if** out is a number **then**
        samples.append(out)
    **end if**
**end while**

---

---

**Algorithm 2** Pseudocode for computing the log pdf of $y$

---

$n \leftarrow$ number of digits after decimal point
nonnum_idxs $\leftarrow$ tokens $\notin$ tokenize(['0', '1', ..., '9', '-', '.', '$\langle t \rangle$'])
full_text $\leftarrow$ prompt + str($y$)
y_idxs $\leftarrow$ indices of the tokens that correspond to y in full_text
logits $\leftarrow$ model(full_text)
y_logits $\leftarrow$ logits[y_idxs]
y_logits[nonnum_idxs] $\leftarrow$ -100
y_logpmf $\leftarrow$ CrossEntropy(logits = y_logits[:-1], targets = str($y$)[1:]).sum( ) ▷ Mass of bin that includes $y$
y_logpdf $\leftarrow$ y_logpmf + $n \log 10$                    ▷ Convert mass to continuous likelihood

---

## C   Sample Prompts

Figure C.1 depicts three observed training points and four target locations. Below are sample prompts for various configurations discussed in the paper. $T$ refers to problem related text.

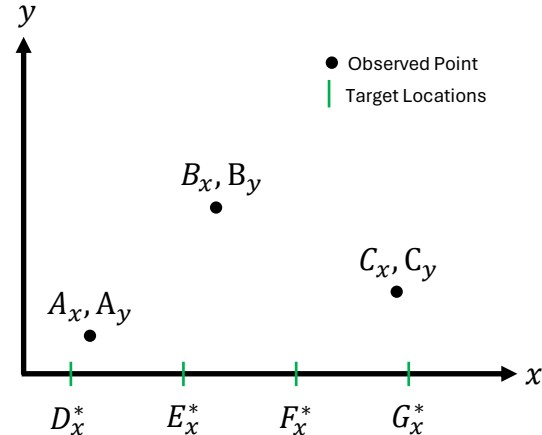

Figure C.1: Three observed training points and four target locations which serve as the basis for the example prompts.

**Independent Marginal Prompts**

*Sequential*:

"$T\langle t\rangle A_x, A_y\langle t\rangle B_x, B_y\langle t\rangle C_x, C_y\langle t\rangle D_x^*$"

"$T\langle t\rangle A_x, A_y\langle t\rangle B_x, B_y\langle t\rangle C_x, C_y\langle t\rangle E_x^*$"

"$T\langle t\rangle A_x, A_y\langle t\rangle B_x, B_y\langle t\rangle C_x, C_y\langle t\rangle F_x^*$"

"$T\langle t\rangle A_x, A_y\langle t\rangle B_x, B_y\langle t\rangle C_x, C_y\langle t\rangle G_x^*$"

*Random*:

"$T\langle t\rangle C_x, C_y\langle t\rangle A_x, A_y\langle t\rangle B_x, B_y\langle t\rangle D_x^*$"

"$T\langle t\rangle C_x, C_y\langle t\rangle A_x, A_y\langle t\rangle B_x, B_y\langle t\rangle E_x^*$"

"$T\langle t\rangle C_x, C_y\langle t\rangle A_x, A_y\langle t\rangle B_x, B_y\langle t\rangle F_x^*$"

"$T\langle t\rangle C_x, C_y\langle t\rangle A_x, A_y\langle t\rangle B_x, B_y\langle t\rangle G_x^*$"

*Distance*:

"$T\langle t\rangle C_x, C_y\langle t\rangle B_x, B_y\langle t\rangle A_x, A_y\langle t\rangle D_x^*$"

"$T\langle t\rangle C_x, C_y\langle t\rangle A_x, A_y\langle t\rangle B_x, B_y\langle t\rangle E_x^*$"

"$T\langle t\rangle A_x, A_y\langle t\rangle C_x, C_y\langle t\rangle B_x, B_y\langle t\rangle F_x^*$"

"$T\langle t\rangle A_x, A_y\langle t\rangle B_x, B_y\langle t\rangle C_x, C_y\langle t\rangle G_x^*$"

**Autoregressive Prompts**

*Sequential*:

"$T\langle t\rangle A_x, A_y\langle t\rangle B_x, B_y\langle t\rangle C_x, C_y\langle t\rangle D_x^*$"

"$T\langle t\rangle A_x, A_y\langle t\rangle B_x, B_y\langle t\rangle C_x, C_y\langle t\rangle D_x^*, D_y^*\langle t\rangle E_x^*$"

"$T\langle t\rangle A_x, A_y\langle t\rangle B_x, B_y\langle t\rangle C_x, C_y\langle t\rangle D_x^*, D_y^*\langle t\rangle E_x^*, E_y^*\langle t\rangle F_x^*$"

"$T\langle t\rangle A_x, A_y\langle t\rangle B_x, B_y\langle t\rangle C_x, C_y\langle t\rangle D_x^*, D_y^*\langle t\rangle E_x^*, E_y^*\langle t\rangle F_x^*, F_y^*\langle t\rangle G_x^*$"

*Random*:

"$T\langle t\rangle C_x, C_y\langle t\rangle A_x, A_y\langle t\rangle B_x, B_y\langle t\rangle D_x^*$"

"$T\langle t\rangle C_x, C_y\langle t\rangle A_x, A_y\langle t\rangle B_x, B_y\langle t\rangle D_x^*, D_y^*\langle t\rangle E_x^*$"

"$T\langle t\rangle C_x, C_y\langle t\rangle A_x, A_y\langle t\rangle B_x, B_y\langle t\rangle D_x^*, D_y^*\langle t\rangle E_x^*, E_y^*\langle t\rangle F_x^*$"

"$T\langle t\rangle C_x, C_y\langle t\rangle A_x, A_y\langle t\rangle B_x, B_y\langle t\rangle D_x^*, D_y^*\langle t\rangle E_x^*, E_y^*\langle t\rangle F_x^*, F_y^*\langle t\rangle G_x^*$"

*Distance*:

"$T\langle t\rangle C_x, C_y\langle t\rangle B_x, B_y\langle t\rangle A_x, A_y\langle t\rangle D_x^*$"

"$T\langle t\rangle C_x, C_y\langle t\rangle D_x^*, D_y^*\langle t\rangle A_x, A_y\langle t\rangle B_x, B_y\langle t\rangle E_x^*$"

"$T\langle t\rangle D_x^*, D_y^*\langle t\rangle A_x, A_y\langle t\rangle E_x^*, E_y^*\langle t\rangle C_x, C_y\langle t\rangle B_x, B_y\langle t\rangle F_x^*$"

"$T\langle t\rangle D_x^*, D_y^*\langle t\rangle A_x, A_y\langle t\rangle E_x^*, E_y^*\langle t\rangle B_x, B_y\langle t\rangle F_x^*, F_y^*\langle t\rangle C_x, C_y\langle t\rangle G_x^*$"

# D  Dataset Details

This section provides details on the various datasets used in the experiments

## D.1  Function Dataset

We use the 12 synthetic function datasets (Linear, Exponential, Sigmoid, Log, Sine, Beat Inference, Linear + Cosine, Linear $\times$ Sine, Gaussian Wave. Sinc, Quadratic, X $\times$ Sine) from Gruver et al. [2] each of which consists of 200 discrete points. We construct 7 datasets each with 10 random seeds for each function with a subset of 5, 10, 15, 20, 25, 50, and 75 randomly training points sampled from the original 200 points. We add Gaussian noise with $\mu = 0$ and $\sigma = 0.05$ to the training points and then round the values to 2 decimal places. Unless otherwise stated, we use 40 equally spaced target points to sample at.

## D.2  Weather Dataset

The dataset was queried from OpenWeather [36] and consists of daily high temperature, precipitation, and wind speed readings for 86 consecutive days from London, UK commencing on December 12, 2023. The data was recorded after the release dates of the Llama-2 and Mixtral-8x7B LLM release dates to avoid any data leakage into the LLM datasets.

For the "Comparison to LLMTime" experiment, We used the first 50 readings of the temperature data for training data and ask LLMTime and LLMPs to predict/forecast the final 36 values. The authors of LLMTime suggest the method can handle missing values by inputting NaN values in their place. Since LLMPs can work with irregularly spaced and missing data, we also compare the methods with a reduced number of randomly spaced training points.

For the "Simultaneous Temperature, Rainfall, and Wind Speed Regression" experiment we used 30 randomly chosen training points within the first 76 points, leaving the last 10 for extrapolation.

# E  Data Leakage

It is likely that LLMs used in our experiments have been exposed during training to some of the real-world data that we use in our experiments which would give it an advantage against other models. However, we feel confident that the LLMs tested were not simply recalling memorized data – note that in all cases the LLMPs produces a full distribution and not just a deterministic value – and we have taken steps in our experiments to mitigate this issue. When synthetic functions or Fashion MNIST data [12] is used, we have altered the original data via subsampling, rescaling and in some cases adding noise to the datapoints. Any data used from the internet was altered from its original form when given to the model. Some datasets (in particular the Weather Dataset described in Appendix D.2), were explicitly chosen to be recorded after the release dates of the LLMs that they were evaluated on.

# F  Additional Implementation Details

PyTorch is used as the basis for all of the experiments, with the exception of the Gaussian Processes baselines that are implemented using the GPyTorch package [37].

The experiments using the Mixtral 8×7B, Mixtral-8×7B-Instruct [7], Llama-2 70B [8], and Llama-3 70B [9] LLMs were run on two NVidia A100 GPUs with 80 GB of memory. The experiments using the Llama-2 7B [8] and Llama-3 8B [9] LLMs were run on one NVidia 3090 GPU with 24 GB of memory. The total compute used in the paper exceeded 600 GPU hours.

No training was done in our LLM experiments, we simply input the prompt to the LLM and ran it forward to get a prediction for a particular target point.

## F.1  Processing Times

Processing times vary as a function of:

- The GPU used.
- The length of the prompt.
- The number of target points queried.
- The number of tokens required to be generated for a particular target point.
- The number of samples taken at each target point.
- Whether independent or autoregressive sampling is used.

Example experiment processing times:

*Basic Scenario*: Table F.1 indicates that the longer the prompt, the longer the computation time for each target point. For independent sampling (I-LLMP), the prompt length is constant and is only a function of the number of training points as each target point is processed independently. For autoregressive sampling (A-LLMP), the prompt length is a function of both the number of training points and the number of target points since each target point is appended to the prompt as it is sampled.

Table F.1: Times to load the LLM into GPU memory, for the LLM to generate all samples at all target points, and to compute the probability distribution over the true target points. All runs used the Llama-2-7B LLM and were executed on an NVIDIA 3090 GPU with 24GB of memory with a batch size of 10. All times are in seconds.

| Function | Model | Load (s) | Sample (s) | Compute Likelihood (s) |
|---|---|---|---|---|
| Quadratic - 10 Training Points, 40 Target Points | I-LLMP | 5 | 81 | 1 |
| Quadratic - 10 Training Points, 40 Target Points | A-LLMP | 5 | 170 | 3 |
| Quadratic - 50 Training Points, 40 Target Points | I-LLMP | 5 | 259 | 4 |
| Quadratic - 50 Training Points, 40 Target Points | A-LLMP | 5 | 354 | 7 |

*1D Synthetic Data Experiments*:

- **LLM**: Mixtral-8×-7B
- **GPU**: 2 × Nvidia A100, 80 GB
- **Parameters**: A-LLMP, 40 target points, 50 samples, log probabilities
- **Tasks**: 12 functions x 3 seeds x 4 sizes
- **Approximate Time**: 19.6 hours

*Black Box Optimization*:

- **LLM**: Llama-2 7B
- **GPU**: 1 × Nvidia A100, 80 GB
- **Parameters**: I-LLMP, 500 target points, 1 sample
- **Tasks**: 6 functions, 100 trials
- **Approximate Time**: 20 hours

*Fashion MNIST Image Reconstruction*:

- **LLM**: Mixtral-8×-7B
- **GPU**: 2 × Nvidia A100, 80 GB
- **Parameters**: I-LLMP, 400 target points, 50 samples
- **Tasks**: 6 images x 2 sizes
- **Approximate Time**: 15 hours

*Simultaneous Temperature, Rainfall, and Wind Speed Regression*

- **LLM**: Llama-3 8B
- **GPU**: 1 × Nvidia 3090, 24 GB
- **Parameters**: A-LLMP, 40 target points, 50 samples
- **Tasks**: 6 functions, 100 trials
- **Approximate Time**: 31 minutes

# G   Additional Configuration Results

## G.1   Comparing Sampling and Logit Based Distributions

We first investigate whether our logit-based method of eliciting distributions (Appendix A.2) match the sampling distribution of the LLM. In order to estimate the true distribution, we obtain 1000 samples from the LLM at each target location, and fit a histogram using the same bins as our logit-based method. Figures G.2 to G.4 show that our method yields a distribution that is visually similar to the one obtained by sampling.

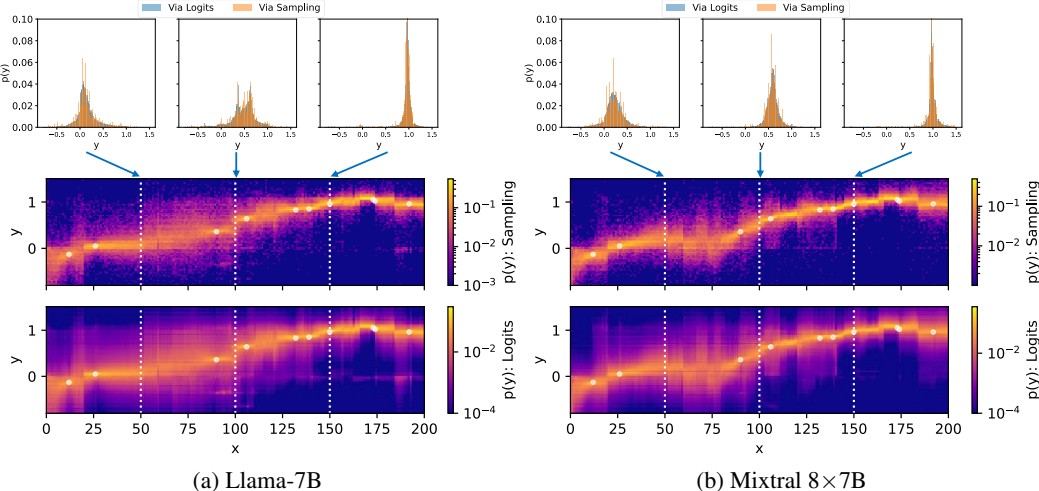

(a) Llama-7B                                                    (b) Mixtral 8×7B

Figure G.2: Visualization of the predictive densities estimated via sampling (*middle*) and model logits (*bottom*) for the Sigmoid function with 10 training points (shown in white). Cross section histograms (*top*) are presented at $x = 50, 100$ and $150$.

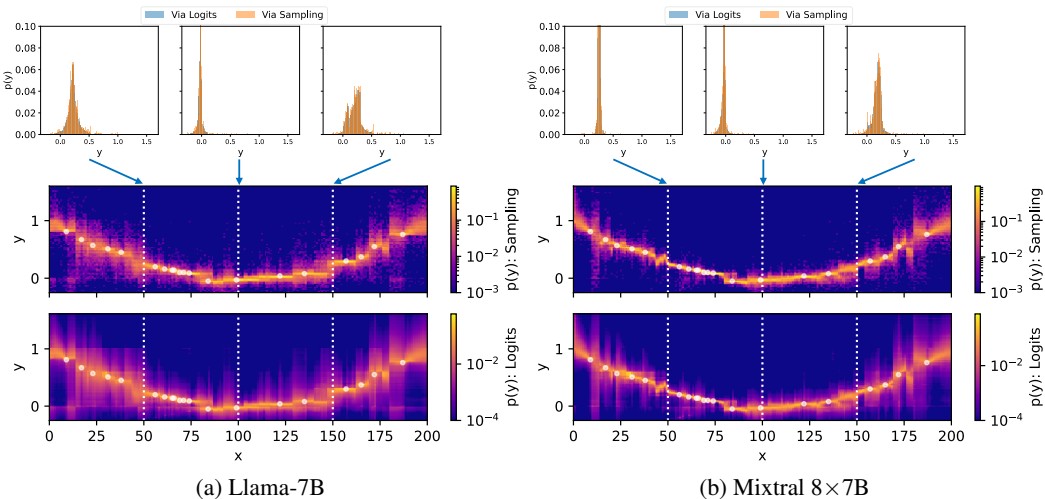

(a) Llama-7B                                                    (b) Mixtral 8×7B

Figure G.3: Visualization of the predictive densities estimated via sampling (*middle*) and model logits (*bottom*) for the Quadratic function with 20 training points (shown in white). Cross section histograms (*top*) are presented at $x = 50, 100$ and $150$.

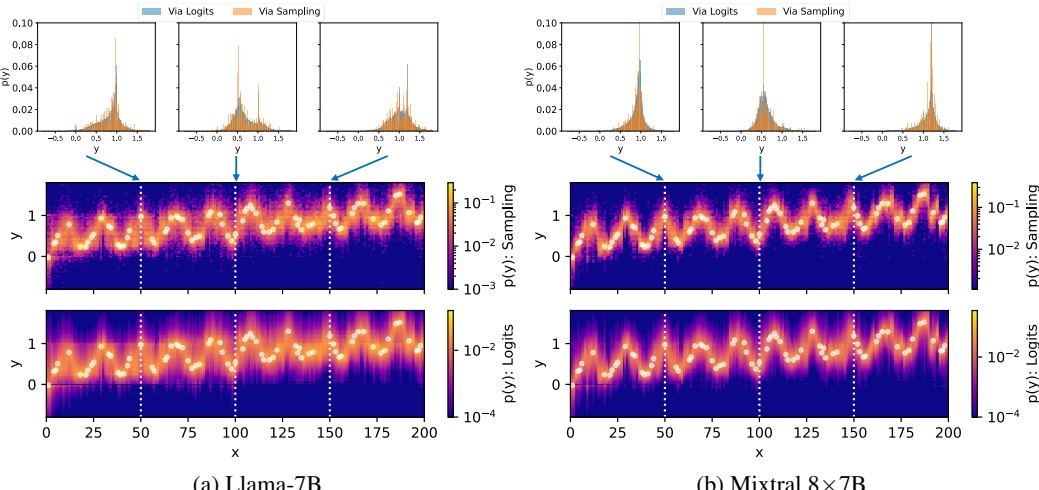

(a) Llama-7B  (b) Mixtral 8×7B

Figure G.4: Visualization of the predictive densities estimated via sampling (*middle*) and model logits (*bottom*) for the Linear + Cosine function with 75 training points (shown in white). Cross section histograms (*top*) are presented at $x = 50, 100$ and $150$.

## G.2 Additional Prompt Format Results

Figure G.5 shows NLL and MAE for various prompt formats and 3 LLMs. Tables G.2 and G.3 show the tabular versions of prompt formatting results.

Overall, LLMPs tested are robust to the prompt format. The results indicate that two separators are required to achieve the best performance. One to separate the $x$ and $y$ values within a pair and another to separate the $x, y$ pairs. The _,_ format uses a comma to separate within a pair and nothing to separate the pairs and it has the worst results. The x_y_ format uses letter prefixes to separate values and pairs with improved metrics. Trading off token efficiency and performance, _,_\n is the best option as it uses only one comma to delimit $x$ and $y$ and \n to delimit $x, y$ pairs. However, given that some regions use a comma as a decimal place, we use _, _\n prompt format in our experiments as it comparable performance and only uses one additional space per pair. The (_, _) and x=_, y=_\n formats are more human readable, but the extra tokens do not improve performance.

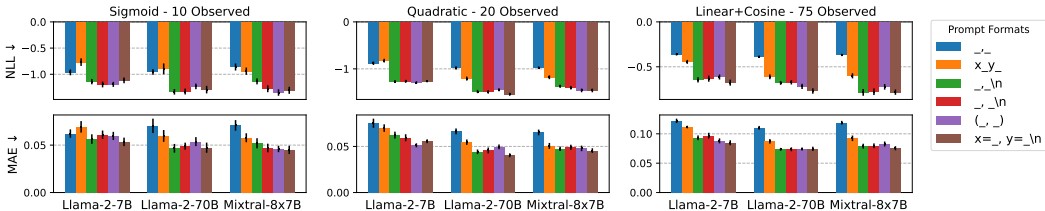

Figure G.5: NLL and MAE for various prompt formats and each LLM. The height of each bar is the mean of 10 random seeds that determine the locations of the observed points. The small black lines at the top of each bar indicates the standard error. The two '_' characters in the legend indicate the positions the $x$ and $y$ values. \n indicates the newline character. From left to right, the prompts are ordered from the most to least token efficient.

Table G.2: NLL for various prompt formats and each LLM. Each entry is the mean and standard error of 10 random seeds that determine the locations of the observed points. From left to right, the prompts are ordered from the most to least token efficient. The number below each function indicates the number of observed points.

| Function | LLM | _,_ | x_y_ | _,_\n | _, _\n | (_, _) | x=_, y=_\n |
|---|---|---|---|---|---|---|---|
| Sigmoid | Llama-2-7B | -0.963±0.056 | -0.768±0.072 | -1.140±0.051 | -1.194±0.055 | -1.192±0.048 | -1.116±0.055 |
| 10 | Llama-2-70B | -0.956±0.053 | -0.897±0.104 | -1.335±0.053 | -1.329±0.056 | -1.231±0.054 | -1.293±0.072 |
| | Mixtral-8x7B | -0.861±0.067 | -0.940±0.069 | -1.135±0.057 | -1.276±0.066 | -1.348±0.062 | -1.306±0.067 |
| Quadratic | Llama-2-7B | -0.882±0.036 | -0.824±0.039 | -1.269±0.032 | -1.266±0.032 | -1.293±0.029 | -1.263±0.023 |
| 20 | Llama-2-70B | -0.980±0.035 | -1.207±0.042 | -1.482±0.034 | -1.489±0.037 | -1.445±0.032 | -1.540±0.032 |
| | Mixtral-8x7B | -0.976±0.028 | -1.179±0.040 | -1.371±0.033 | -1.401±0.038 | -1.459±0.039 | -1.459±0.039 |
| Linear + | Llama-2-7B | -0.362±0.012 | -0.445±0.022 | -0.645±0.029 | -0.632±0.034 | -0.613±0.028 | -0.676±0.033 |
| Cosine | Llama-2-70B | -0.386±0.012 | -0.611±0.027 | -0.679±0.021 | -0.673±0.024 | -0.718±0.029 | -0.769±0.030 |
| 75 | Mixtral-8x7B | -0.368±0.013 | -0.600±0.029 | -0.785±0.038 | -0.778±0.036 | -0.723±0.031 | -0.782±0.030 |

Table G.3: Mean Average Error (MAE) for various prompt formats and each LLM. Each entry is the mean and standard error of 10 random seeds that determine the locations of the observed points. From left to right, the prompts are ordered from the most to least token efficient. The number below each function indicates the number of observed points.

| Function | LLM | _,_ | x_y_ | _,_\n | _, _\n | (_, _) | x=_, y=_\n |
|---|---|---|---|---|---|---|---|
| Sigmoid 10 | Llama-2-7B | 0.062±0.004 | 0.069±0.006 | 0.056±0.005 | 0.061±0.004 | 0.060±0.004 | 0.053±0.004 |
| | Llama-2-70B | 0.070±0.008 | 0.060±0.006 | 0.047±0.005 | 0.049±0.004 | 0.054±0.005 | 0.047±0.005 |
| | Mixtral-8x7B | 0.071±0.006 | 0.058±0.005 | 0.052±0.005 | 0.047±0.005 | 0.046±0.003 | 0.045±0.004 |
| Quadratic 20 | Llama-2-7B | 0.075±0.005 | 0.070±0.004 | 0.062±0.004 | 0.059±0.004 | 0.051±0.002 | 0.056±0.002 |
| | Llama-2-70B | 0.066±0.003 | 0.055±0.003 | 0.044±0.002 | 0.046±0.003 | 0.050±0.003 | 0.040±0.002 |
| | Mixtral-8x7B | 0.065±0.003 | 0.051±0.003 | 0.047±0.002 | 0.049±0.003 | 0.048±0.003 | 0.045±0.003 |
| Linear + Cosine 75 | Llama-2-7B | 0.122±0.004 | 0.112±0.002 | 0.093±0.004 | 0.097±0.005 | 0.088±0.004 | 0.085±0.004 |
| | Llama-2-70B | 0.110±0.003 | 0.087±0.004 | 0.074±0.002 | 0.074±0.003 | 0.074±0.003 | 0.074±0.004 |
| | Mixtral-8x7B | 0.119±0.003 | 0.092±0.005 | 0.079±0.004 | 0.080±0.004 | 0.083±0.004 | 0.075±0.004 |

### G.3 Additional Prompt Ordering Results

We consider the effect of three different orderings of the training data $D_{\text{train}}$ in the prompt:

- *Sequential*: $(x_i, y_i), \in D_{\text{train}}$ are ordered sequentially from smallest to largest $x_i$, regardless of the location of the target point.
- *Random*: $(x_i, y_i), \in D_{\text{train}}$ are randomly ordered.
- *Distance*: For the prediction at target point $x^*$, the training points $(x_i, y_i), \in D_{\text{train}}$ are ordered from largest to smallest distance to the query point $x^*$ i.e. $|x_n^* - x_i|_2$ such that the training points closer to $x^*$ appear later in the prompt.

Figure G.6 shows NLL and MAE for various prompt orderings and each LLM. Table G.4 shows the tabular version of the results.

Distance ordering consistently yields the best results overall. We posit that distance ordering is effective as it provides a hint to the LLM to weigh the contribution of closer points to the current target point to a greater degree. Unless otherwise noted, we use distance ordering for our experiments.

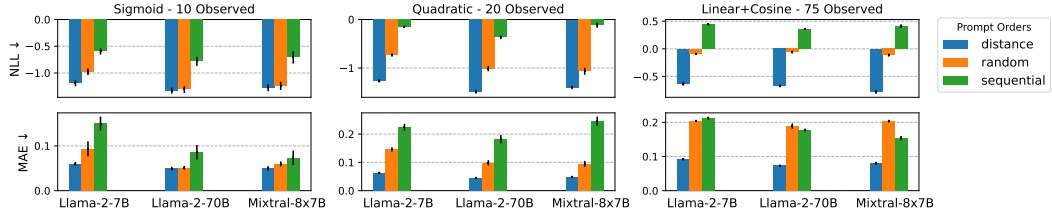

Figure G.6: NLL and MAE for various prompt orderings and each LLM. The height of each bar is the mean of 10 random seeds that determine the locations of the observed points. The small black lines at the top of each bar indicates the standard error.

Table G.4: Mean Average Error (MAE) and NLL for various prompt orderings and each LLM. Each entry is the mean and standard error of 10 random seeds that determine the locations of the observed points. The number below each function indicates the number of observed points.

| Function | LLM | Distance MAE ↓ | Distance NLL ↓ | Random MAE ↓ | Random NLL ↓ | Sequential MAE ↓ | Sequential NLL ↓ |
|---|---|---|---|---|---|---|---|
| Sigmoid 10 | Llama-2-7B | 0.060±0.004 | -1.194±0.055 | 0.093±0.017 | -0.977±0.063 | 0.150±0.016 | -0.597±0.059 |
| | Llama-2-70B | 0.049±0.004 | -1.329±0.056 | 0.051±0.004 | -1.307±0.066 | 0.086±0.016 | -0.782±0.085 |
| | Mixtral-8x7B | 0.050±0.005 | -1.276±0.066 | 0.060±0.006 | -1.240±0.077 | 0.073±0.016 | -0.707±0.116 |
| Quadratic 20 | Llama-2-7B | 0.063±0.004 | -1.266±0.032 | 0.146±0.007 | -0.731±0.034 | 0.224±0.012 | -0.147±0.019 |
| | Llama-2-70B | 0.046±0.003 | -1.490±0.037 | 0.099±0.009 | -1.013±0.055 | 0.182±0.014 | -0.368±0.035 |
| | Mixtral-8x7B | 0.049±0.003 | -1.401±0.038 | 0.095±0.011 | -1.066±0.074 | 0.246±0.016 | -0.117±0.053 |
| Linear + Cosine 75 | Llama-2-7B | 0.092±0.003 | -0.632±0.034 | 0.205±0.003 | -0.086±0.015 | 0.213±0.004 | 0.445±0.022 |
| | Llama-2-70B | 0.074±0.003 | -0.673±0.024 | 0.189±0.008 | -0.058±0.025 | 0.178±0.004 | 0.361±0.018 |
| | Mixtral-8x7B | 0.080±0.004 | -0.778±0.036 | 0.204±0.004 | -0.114±0.027 | 0.154±0.006 | 0.410±0.034 |

## G.4    Additional Prompt $y$-Scaling Results

In this experiment, we examine the effect of the magnitude and sign of the $y$-values of the task given to the LLM when no other contextual information is provided. We take the same three synthetic examples but scale the $y$-values to be in the ranges $[0, 1]$, $[-1, 1]$, $[0, 10]$ and $[-1000, 1000]$.

Figure G.7 shows NLL and MAE for various prompt $y$-scaling and each LLM. Table G.5 shows the tabular results. The raw values given to the LLM are scaled meaning the observation noise is scaled accordingly. We have scaled the likelihoods and MAE values to compensate for the difference in range. According to the evaluation metrics we observe that performance degrades with increased range and incorporating negative values also hurts MAE. This is due to the fact that when the range is wider, the LLM must accurately generate more numerical digits and potentially a negative sign when predicting $f(x^*)$.

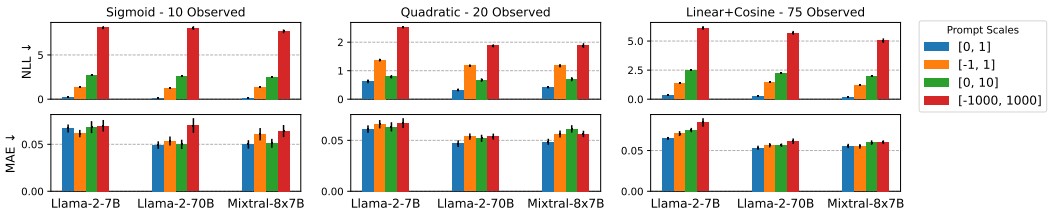

Figure G.7: NLL and MAE for various prompt y-scalings and each LLM. The height of each bar is the mean of 10 random seeds that determine the locations of the observed points. The small black lines at the top of each bar indicates the standard error.

Table G.5: MAE and NLL for various $y$-scaling ranges and three LLMs. Each entry is the mean and standard error of 10 random seeds that determine the locations of the observed points. The number below each function indicates the number of observed points.

| Function | LLM | [0,1] | | [-1,1] | | [0,10] | | [-1000, 1000] | |
| | | MAE ↓ | NLL ↓ | MAE ↓ | NLL ↓ | MAE ↓ | NLL ↓ | MAE ↓ | NLL ↓ |
|---|---|---|---|---|---|---|---|---|---|
| Sigmoid | Llama-2-7B | 0.067 +/- 0.004 | 0.212 +/- 0.053 | 0.061 +/- 0.004 | 1.327 +/- 0.057 | 0.068 +/- 0.006 | 2.701 +/- 0.075 | 0.070 +/- 0.006 | 8.087 +/- 0.173 |
| 10 | Llama-2-70B | 0.049 +/- 0.004 | 0.086 +/- 0.049 | 0.054 +/- 0.005 | 1.246 +/- 0.066 | 0.050 +/- 0.005 | 2.565 +/- 0.062 | 0.070 +/- 0.008 | 8.036 +/- 0.210 |
| | Mixtral-8x7B | 0.050 +/- 0.004 | 0.120 +/- 0.065 | 0.061 +/- 0.007 | 1.343 +/- 0.065 | 0.051 +/- 0.005 | 2.502 +/- 0.085 | 0.064 +/- 0.006 | 7.668 +/- 0.212 |
| Quadratic | Llama-2-7B | 0.061 +/- 0.004 | 0.624 +/- 0.066 | 0.066 +/- 0.004 | 1.372 +/- 0.048 | 0.063 +/- 0.005 | 0.788 +/- 0.061 | 0.067 +/- 0.005 | 2.524 +/- 0.041 |
| 20 | Llama-2-70B | 0.047 +/- 0.003 | 0.324 +/- 0.049 | 0.054 +/- 0.003 | 1.176 +/- 0.047 | 0.052 +/- 0.003 | 0.669 +/- 0.063 | 0.054 +/- 0.003 | 1.874 +/- 0.052 |
| | Mixtral-8x7B | 0.049 +/- 0.003 | 0.417 +/- 0.040 | 0.056 +/- 0.003 | 1.175 +/- 0.059 | 0.061 +/- 0.004 | 0.702 +/- 0.072 | 0.056 +/- 0.003 | 1.883 +/- 0.082 |
| Linear + | Llama-2-7B | 0.065 +/- 0.002 | 0.339 +/- 0.032 | 0.071 +/- 0.003 | 1.374 +/- 0.036 | 0.075 +/- 0.003 | 2.513 +/- 0.034 | 0.084 +/- 0.005 | 6.130 +/- 0.156 |
| Cosine | Llama-2-70B | 0.053 +/- 0.003 | 0.276 +/- 0.039 | 0.056 +/- 0.003 | 1.453 +/- 0.033 | 0.057 +/- 0.002 | 2.245 +/- 0.041 | 0.061 +/- 0.003 | 5.709 +/- 0.163 |
| 75 | Mixtral-8x7B | 0.056 +/- 0.003 | 0.193 +/- 0.036 | 0.055 +/- 0.003 | 1.199 +/- 0.035 | 0.060 +/- 0.003 | 1.999 +/- 0.066 | 0.060 +/- 0.002 | 5.036 +/- 0.196 |

However, observing the plots in Figure G.8 of the predictive distribution on each scale, the model gives reasonable predictions regardless of scale. If no scenario context is provided via text to the LLM, rescaling task values to be approximately between $0$ and $1$ improves performance in our experiments. However, in general we use unscaled data so that we can examine the prior beliefs learned by the LLM about tasks communicated through the raw values.

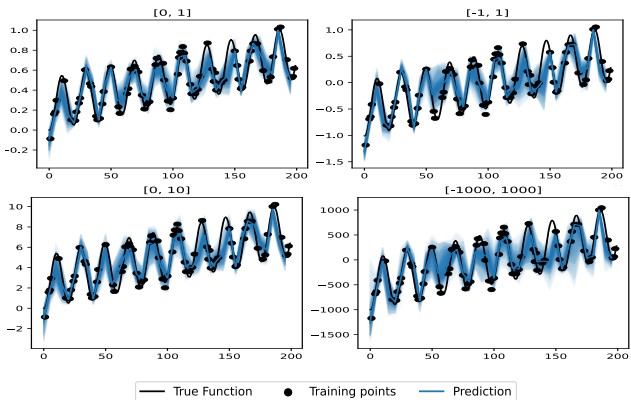

Figure G.8: Predictive distributions given by the Mixtral-8×7B LLM on scaled Linear + Cos with 75 observations. This example exhibited one of the largest variation in metrics as a result of scaling. Despite this, all predictive distributions look reasonable.

## G.5 top-$p$ and temperature results

Figure G.9 shows how MAE varies with LLM top-$p$ and temperature. Table G.6 shows the tabular version of the results.

Surprisingly, all LLM's are insensitive to temperature and top-$p$ with respect to MAE.

Though not evident from these MAE results, we sometimes observed that using a top-$p$ of 1.0 can result in some extreme values in samples. However, we consider temperature = 1.0, and top-$p$ = 1.0 closest to the default distribution given by the LLM. Since it had competitive performance with the other options, we use these settings to compute log-likelihoods in our experiments which allows us to examine the default characteristics of the LLM's predictive distribution.

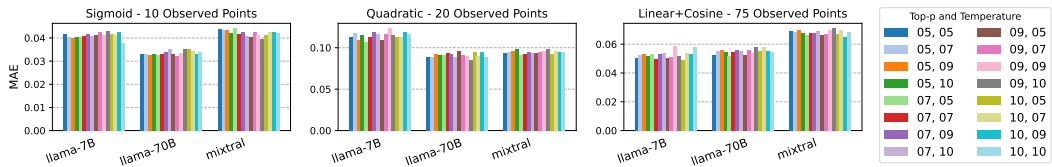

Figure G.9: MAE (lower is better) for various temperature and top-$p$ settings and each LLM. All LLM's are relatively insensitive to temperature and top p with respect to MAE.

Table G.6: MAE (lower is better) for various top-$p$ and temperature settings and all LLMs.

| Function | LLM | Temperature = 0.5 | | | | Temperature = 0.7 | | | | Temperature = 0.9 | | | | Temperature = 1.0 | | | |
|---|---|---|---|---|---|---|---|---|---|---|---|---|---|---|---|---|---|
| | | p=0.5 | p=0.7 | p=0.9 | p=1.0 | p=0.5 | p=0.7 | p=0.9 | p=1.0 | p=0.5 | p=0.7 | p=0.9 | p=1.0 | p=0.5 | p=0.7 | p=0.9 | p=1.0 |
| Sigmoid | L-7B | 0.0329 | 0.033 | 0.0328 | 0.0329 | 0.0328 | 0.0331 | 0.0337 | 0.0351 | 0.0331 | 0.0322 | 0.0334 | 0.035 | 0.035 | 0.0345 | 0.0331 | 0.0339 |
| | Mix | 0.0439 | 0.0436 | 0.0434 | 0.042 | 0.0441 | 0.0419 | 0.0427 | 0.0406 | 0.0404 | 0.0426 | 0.0412 | 0.0394 | 0.0414 | 0.0425 | 0.0426 | 0.0421 |
| | L-70B | 0.045 | 0.0446 | 0.0439 | 0.0429 | 0.0459 | 0.0429 | 0.0417 | 0.0407 | 0.0459 | 0.0396 | 0.0409 | 0.0422 | 0.0452 | 0.0429 | 0.041 | 0.041 |
| Square | L-7B | 0.089 | 0.0886 | 0.0918 | 0.0906 | 0.091 | 0.0931 | 0.0926 | 0.089 | 0.0955 | 0.0911 | 0.0899 | 0.0846 | 0.0951 | 0.09 | 0.0941 | 0.0888 |
| | Mix | 0.094 | 0.0952 | 0.0961 | 0.0986 | 0.0914 | 0.0919 | 0.0945 | 0.094 | 0.0938 | 0.0951 | 0.0954 | 0.0982 | 0.092 | 0.0958 | 0.0941 | 0.0942 |
| | L-70B | 0.1031 | 0.0991 | 0.1031 | 0.1077 | 0.1011 | 0.1015 | 0.1067 | 0.1052 | 0.1025 | 0.1066 | 0.1082 | 0.1066 | 0.1059 | 0.1071 | 0.1104 | 0.1152 |
| Linear + Cosine | L-7B | 0.0524 | 0.0554 | 0.056 | 0.0544 | 0.052 | 0.0546 | 0.0561 | 0.0551 | 0.0525 | 0.0561 | 0.0541 | 0.0583 | 0.0553 | 0.058 | 0.055 | 0.0544 |
| | Mix | 0.0691 | 0.0686 | 0.0696 | 0.0674 | 0.0662 | 0.0674 | 0.0674 | 0.0689 | 0.0664 | 0.0671 | 0.07 | 0.0709 | 0.0671 | 0.0699 | 0.0648 | 0.0685 |
| | L-70B | 0.0661 | 0.0645 | 0.0713 | 0.0701 | 0.0669 | 0.0681 | 0.0728 | 0.075 | 0.0662 | 0.0729 | 0.0781 | 0.0785 | 0.0709 | 0.0703 | 0.0826 | 0.0805 |

## G.6 Additional Autoregressive Sampling Results

Figure G.10 shows NLL and MAE of random and distance training point orderings for A-LLMP and I-LLMP and each LLM. Table G.7 shows the tabular results.

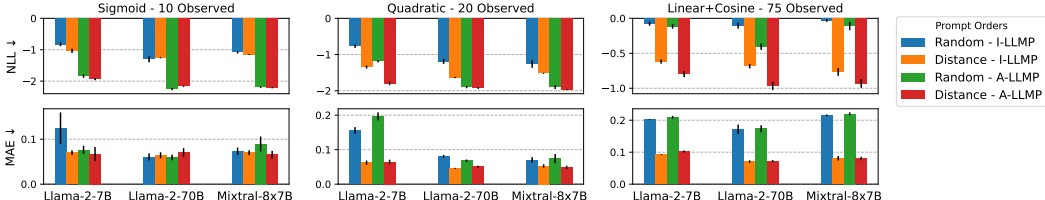

Figure G.10: NLL and MAE for various prompt y-scalings and each LLM. The height of each bar is the mean of 3 random seeds that determine the locations of the observed points. The small black lines at the top of each bar indicates the standard error.

Table G.7: Mean Average Error (MAE) and Negative Log Likelihood (NLL) for autoregressive and marginal sampling with two different prompt orderings and three LLMs.

| Function | LLM | Random IND-LLMP | | Distance IND-LLMP | | Random AUTO-LLMP | | Distance AUTO-LLMP | |
|---|---|---|---|---|---|---|---|---|---|
| | | MAE ↓ | NLL ↓ | MAE ↓ | NLL ↓ | MAE ↓ | NLL ↓ | MAE ↓ | NLL ↓ |
| Sigmoid 10 | Llama-2-7B | 0.125±0.035 | -0.829±0.061 | 0.070±0.005 | -1.035±0.070 | 0.076±0.009 | -1.843±0.052 | 0.067±0.016 | -1.940±0.031 |
| | Llama-2-70B | 0.061±0.008 | -1.303±0.098 | 0.064±0.007 | -1.257±0.016 | 0.060±0.006 | -2.252±0.034 | 0.070±0.010 | -2.162±0.019 |
| | Mixtral-8x7B | 0.073±0.008 | -1.082±0.040 | 0.070±0.005 | -1.153±0.012 | 0.089±0.017 | -2.196±0.023 | 0.065±0.009 | -2.217±0.012 |
| Quadratic 20 | Llama-2-7B | 0.156±0.010 | -0.769±0.044 | 0.062±0.006 | -1.347±0.042 | 0.196±0.012 | -1.184±0.030 | 0.064±0.007 | -1.795±0.049 |
| | Llama-2-70B | 0.081±0.004 | -1.190±0.069 | 0.046±0.001 | -1.634±0.018 | 0.068±0.004 | -1.897±0.034 | 0.051±0.003 | -1.924±0.018 |
| | Mixtral-8x7B | 0.070±0.008 | -1.261±0.103 | 0.053±0.005 | -1.514±0.008 | 0.074±0.013 | -1.900±0.054 | 0.049±0.005 | -1.970±0.013 |
| Linear + Cosine 75 | Llama-2-7B | 0.203±0.001 | -0.076±0.030 | 0.093±0.001 | -0.618±0.031 | 0.209±0.005 | -0.116±0.031 | 0.102±0.003 | -0.799±0.042 |
| | Llama-2-70B | 0.172±0.015 | -0.104±0.043 | 0.070±0.004 | -0.685±0.031 | 0.173±0.011 | -0.405±0.046 | 0.072±0.004 | -0.968±0.058 |
| | Mixtral-8x7B | 0.215±0.003 | -0.030±0.020 | 0.081±0.007 | -0.766±0.056 | 0.220±0.005 | -0.111±0.059 | 0.080±0.006 | -0.931±0.063 |

## G.7 Additional Autoregressive Process Results

Figure G.11 shows the MAE results for the autoregressive process experiments. Figures G.12 and G.13 show the Avg log $p(y)$ and MAE for 10 different orderings of the query points.

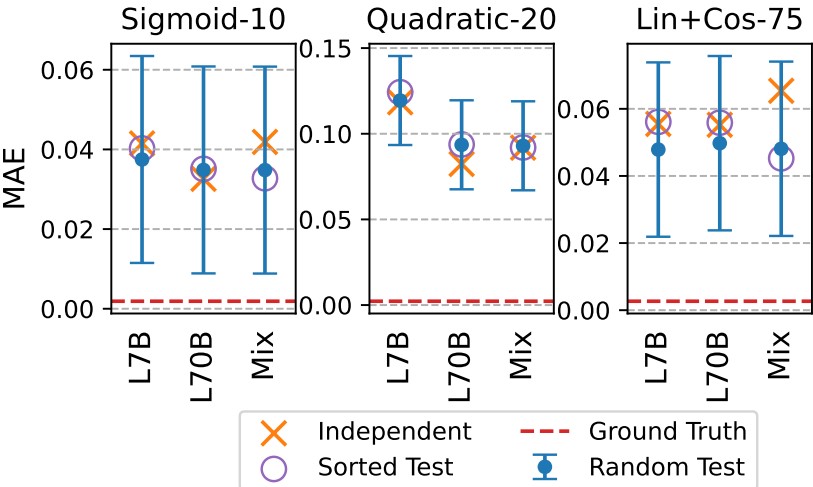

Figure G.11: Autoregressive process MAE results.

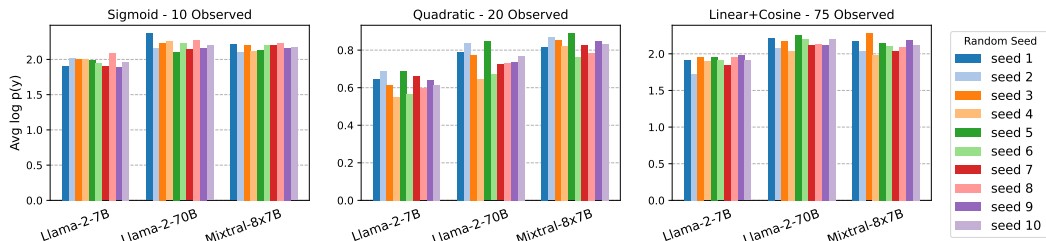

Figure G.12: Avg log $p(y)$ for the 10 seeds for each LLM for the autoregressive process experiment.

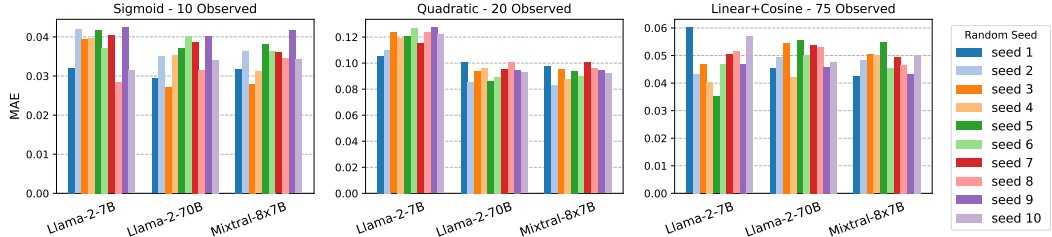

Figure G.13: MAE for the 10 seeds for each LLM for the autoregressive process experiment.

# H Additional LLMP Performance Details and Results

## H.1 Additional Comparison to Gaussian Processes (GP) Results

Figures H.14 to H.25 shows regression results from the Mixtral-8×7B LLM and an RBF kernel GP for the 12 different synthetic functions.

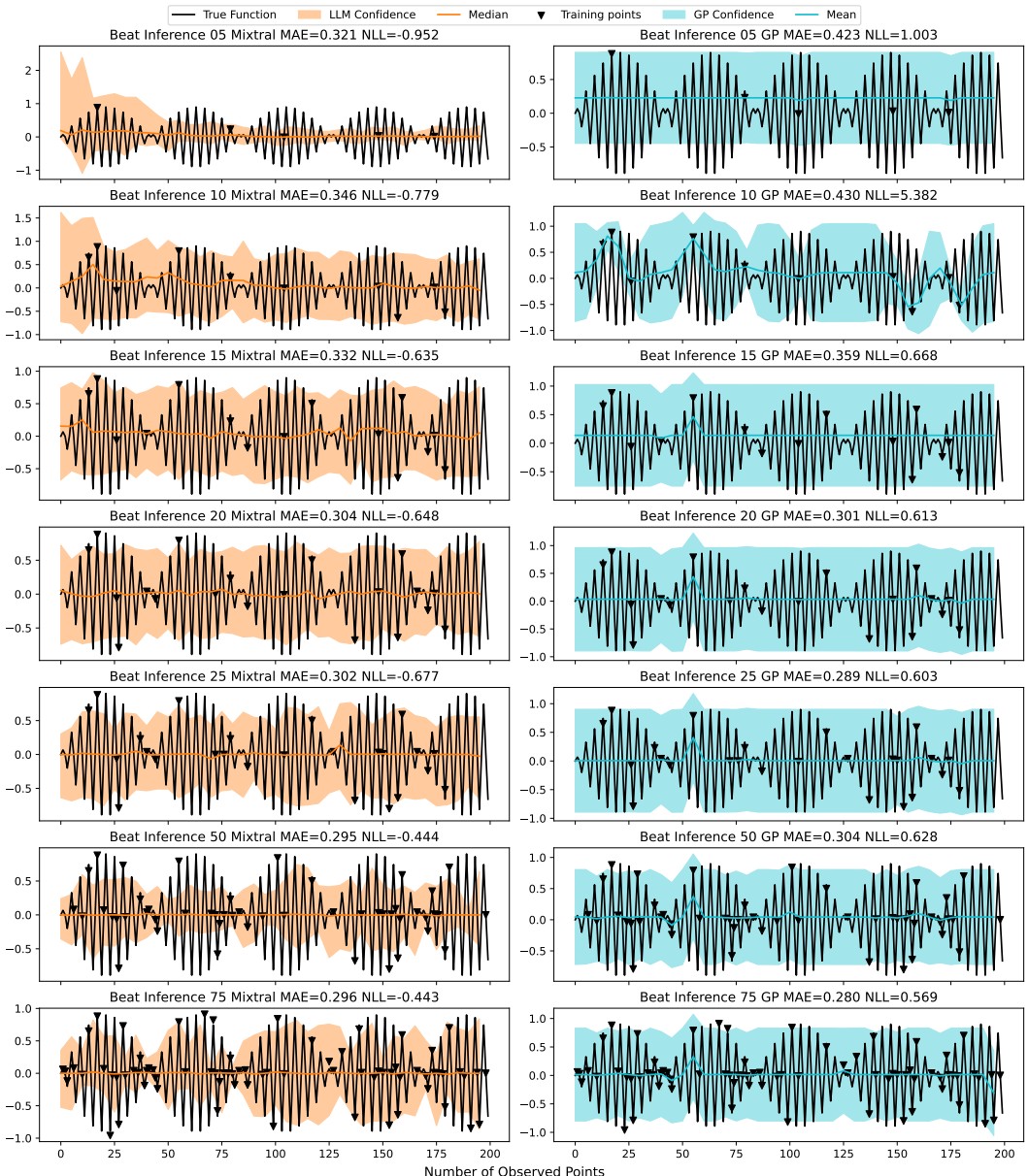

Figure H.14: MAE (lower is better) and NLL (lower is better) for the Mixtral-8×7B LLM versus a GP as a function of the number of observed points for the Beat function. The GP uses an RBF kernel with optimized length scale and noise.

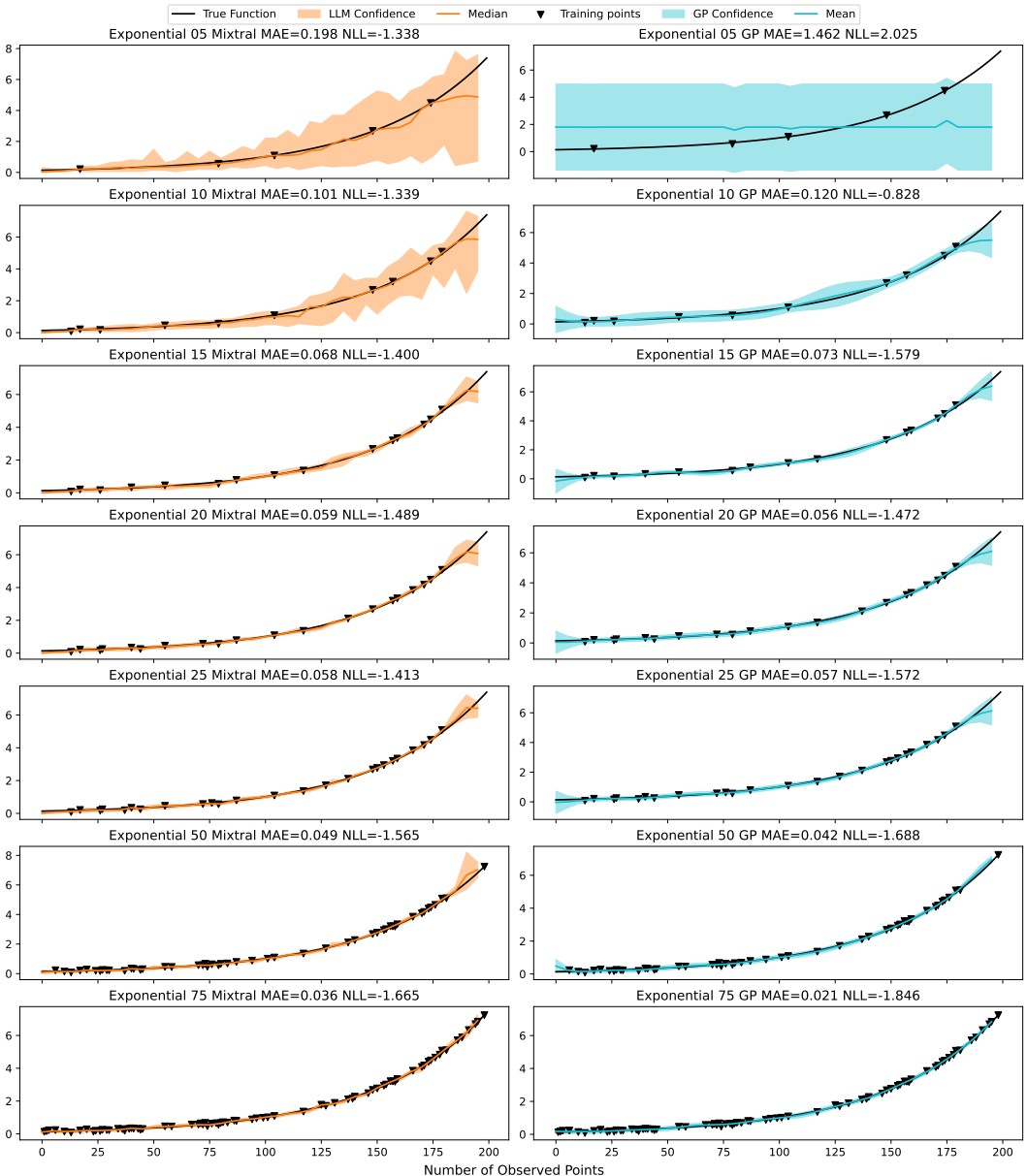

Figure H.15: MAE (lower is better) and NLL (lower is better) for the Mixtral-8×7B LLM versus a GP as a function of the number of observed points for the Exponential function. The GP uses an RBF kernel with optimized length scale and noise.

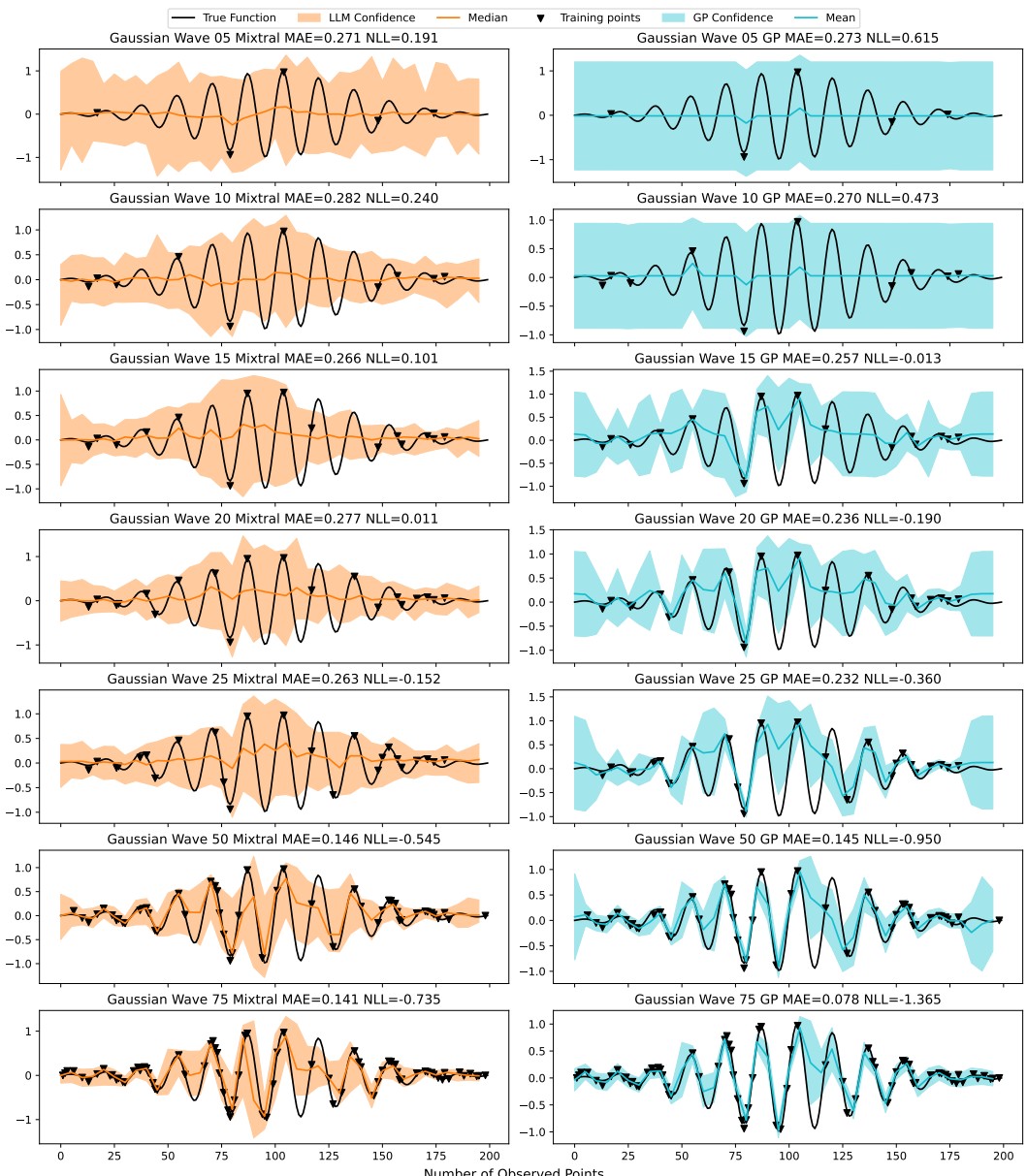

Figure H.16: MAE (lower is better) and NLL (lower is better) for the Mixtral-8×7B LLM versus a GP as a function of the number of observed points for the Gaussian Wave function. The GP uses an RBF kernel with optimized length scale and noise.

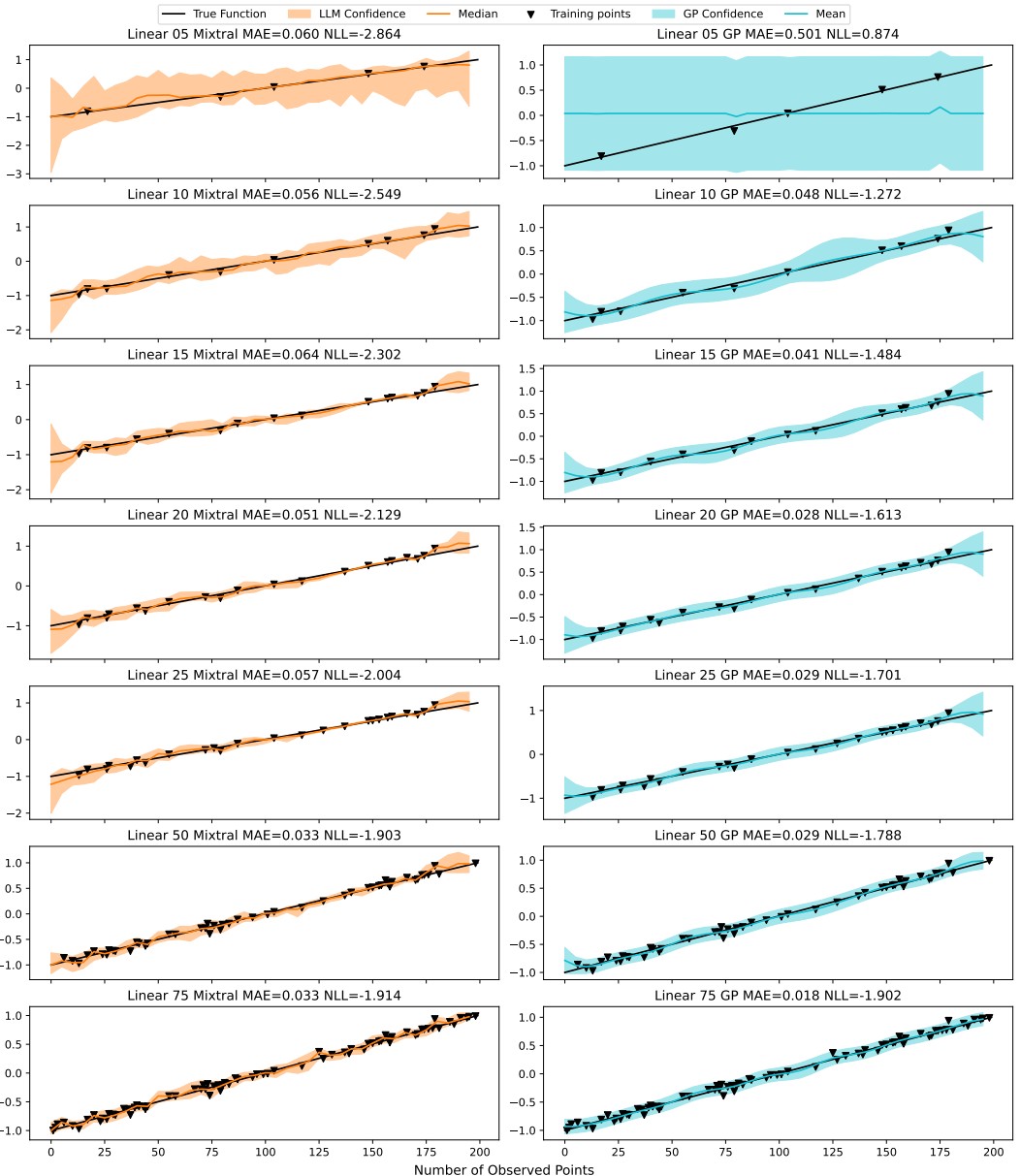

Figure H.17: MAE (lower is better) and NLL (lower is better) for the Mixtral-8×7B LLM versus a GP as a function of the number of observed points for the Linear function. The GP uses an RBF kernel with optimized length scale and noise.

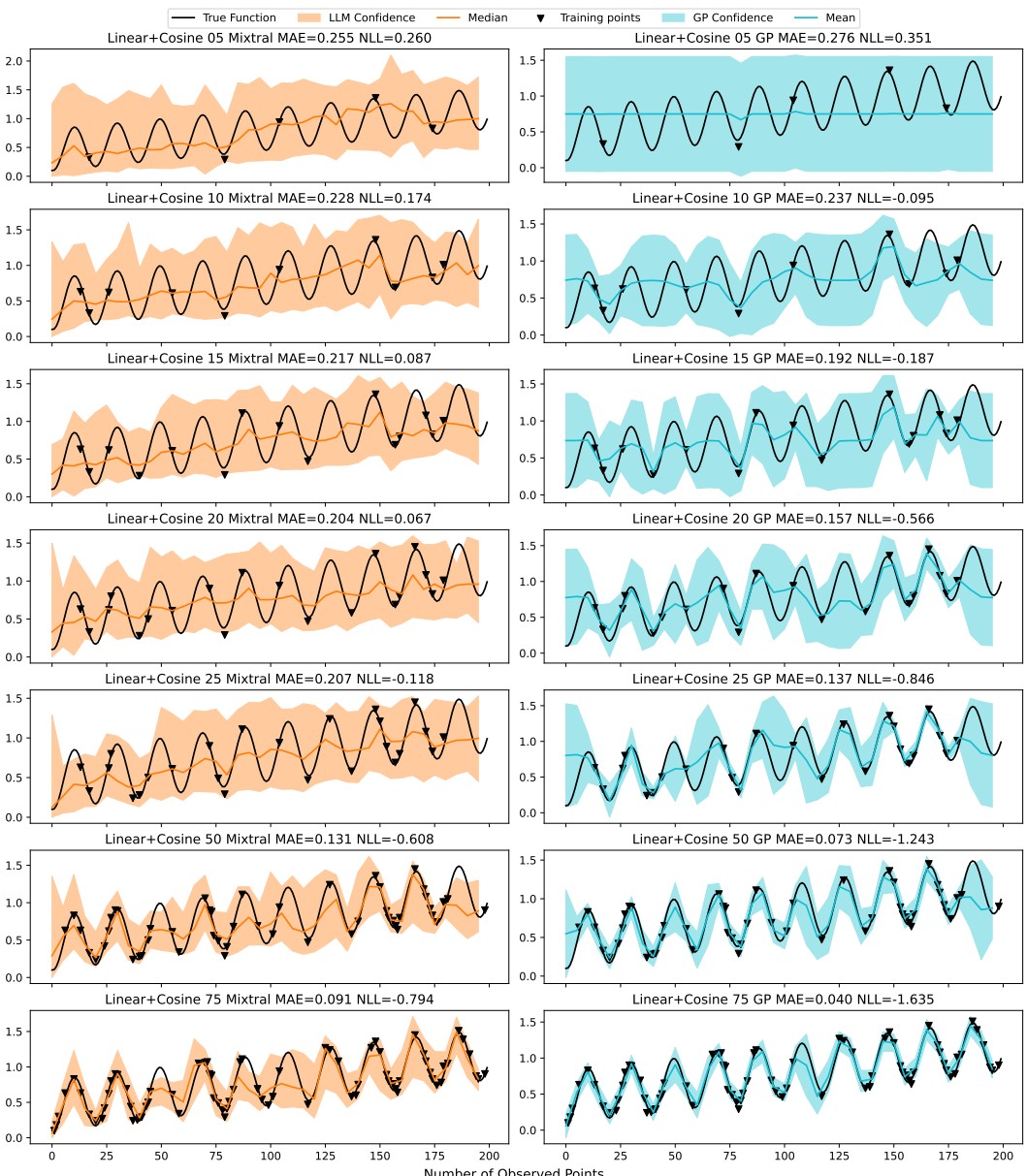

Figure H.18: MAE (lower is better) and NLL (lower is better) for the Mixtral-8×7B LLM versus a GP as a function of the number of observed points for the Linear + Cosine function. The GP uses an RBF kernel with optimized length scale and noise.

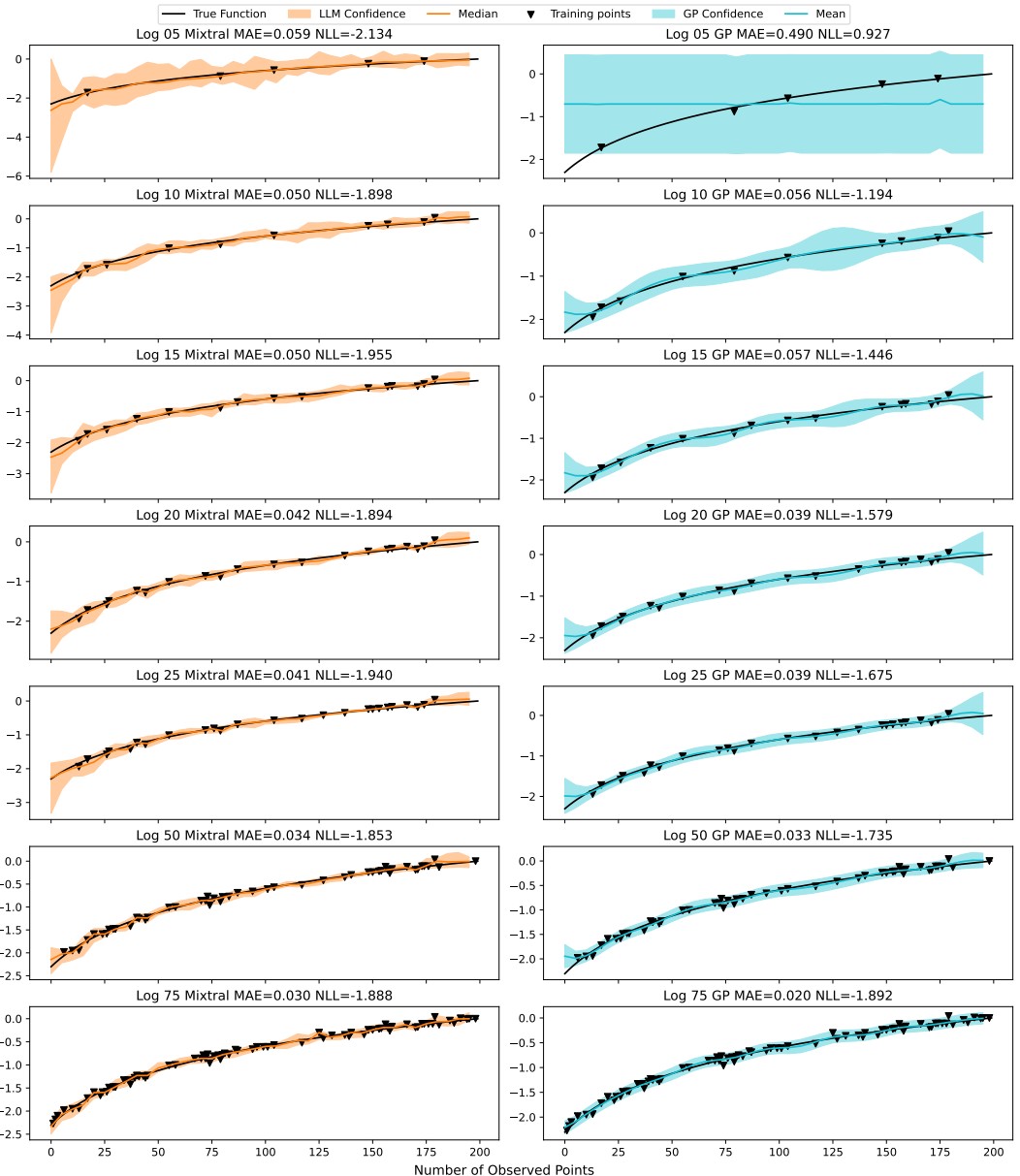

Figure H.19: MAE (lower is better) and NLL (lower is better) for the Mixtral-8×7B LLM versus a GP as a function of the number of observed points for the Log function. The GP uses an RBF kernel with optimized length scale and noise.

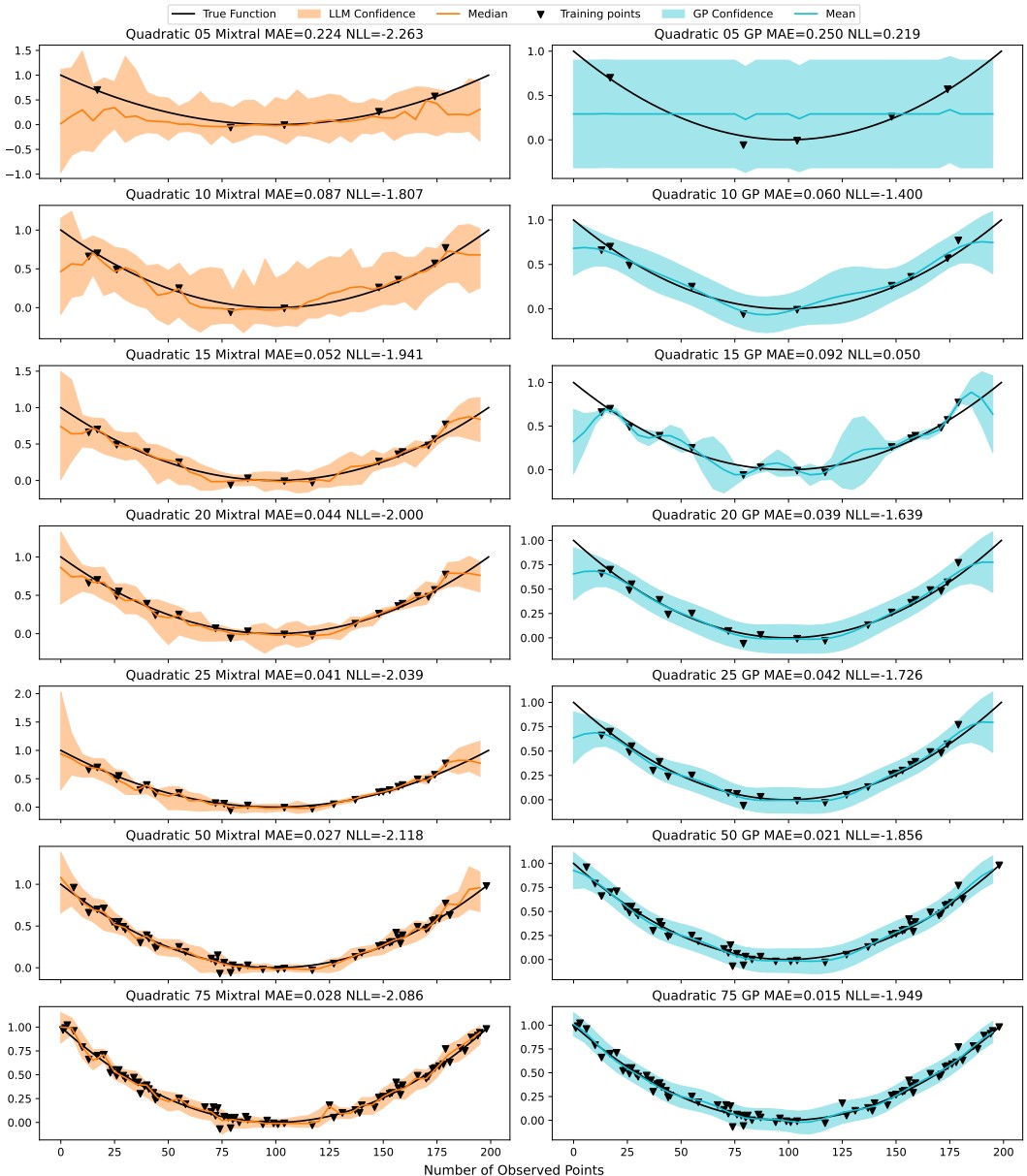

Figure H.20: MAE (lower is better) and NLL (lower is better) for the Mixtral-8×7B LLM versus a GP as a function of the number of observed points for the Quadratic function. The GP uses an RBF kernel with optimized length scale and noise.

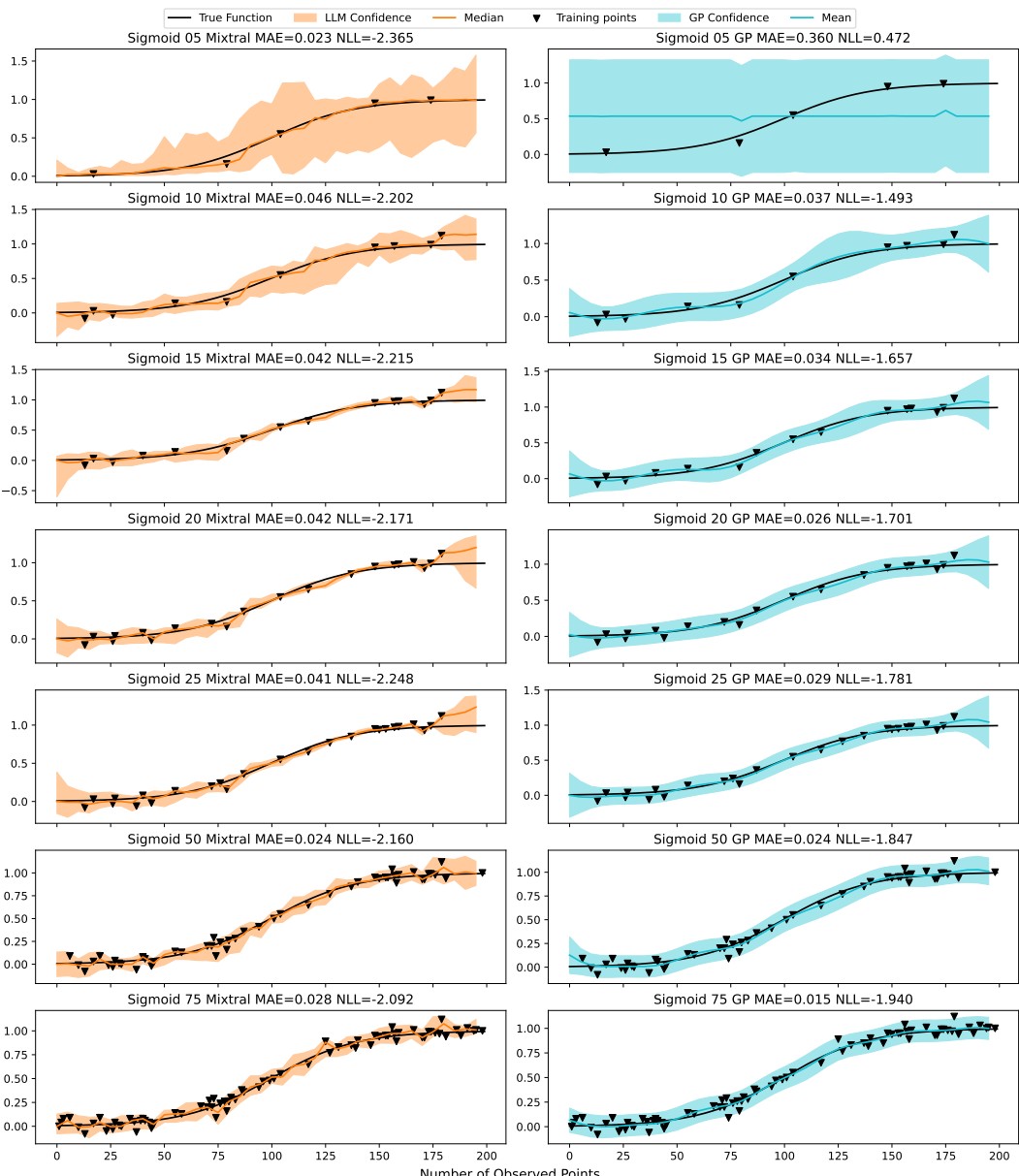

Figure H.21: MAE (lower is better) and NLL (lower is better) for the Mixtral-8×7B LLM versus a GP as a function of the number of observed points for the Sigmoid function. The GP uses an RBF kernel with optimized length scale and noise.

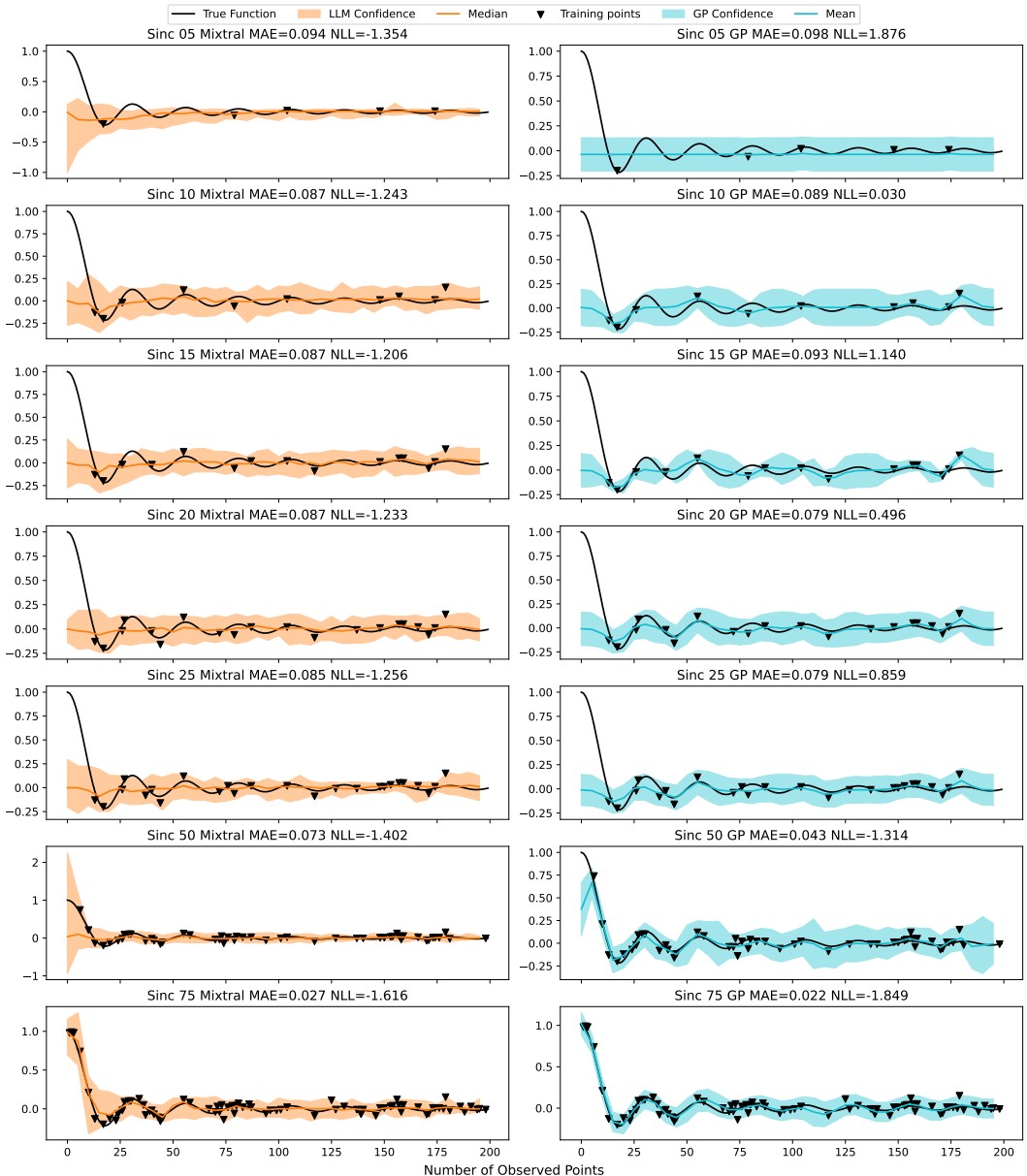

Figure H.22: MAE (lower is better) and NLL (lower is better) for the Mixtral-8×7B LLM versus a GP as a function of the number of observed points for the Sinc function. The GP uses an RBF kernel with optimized length scale and noise.

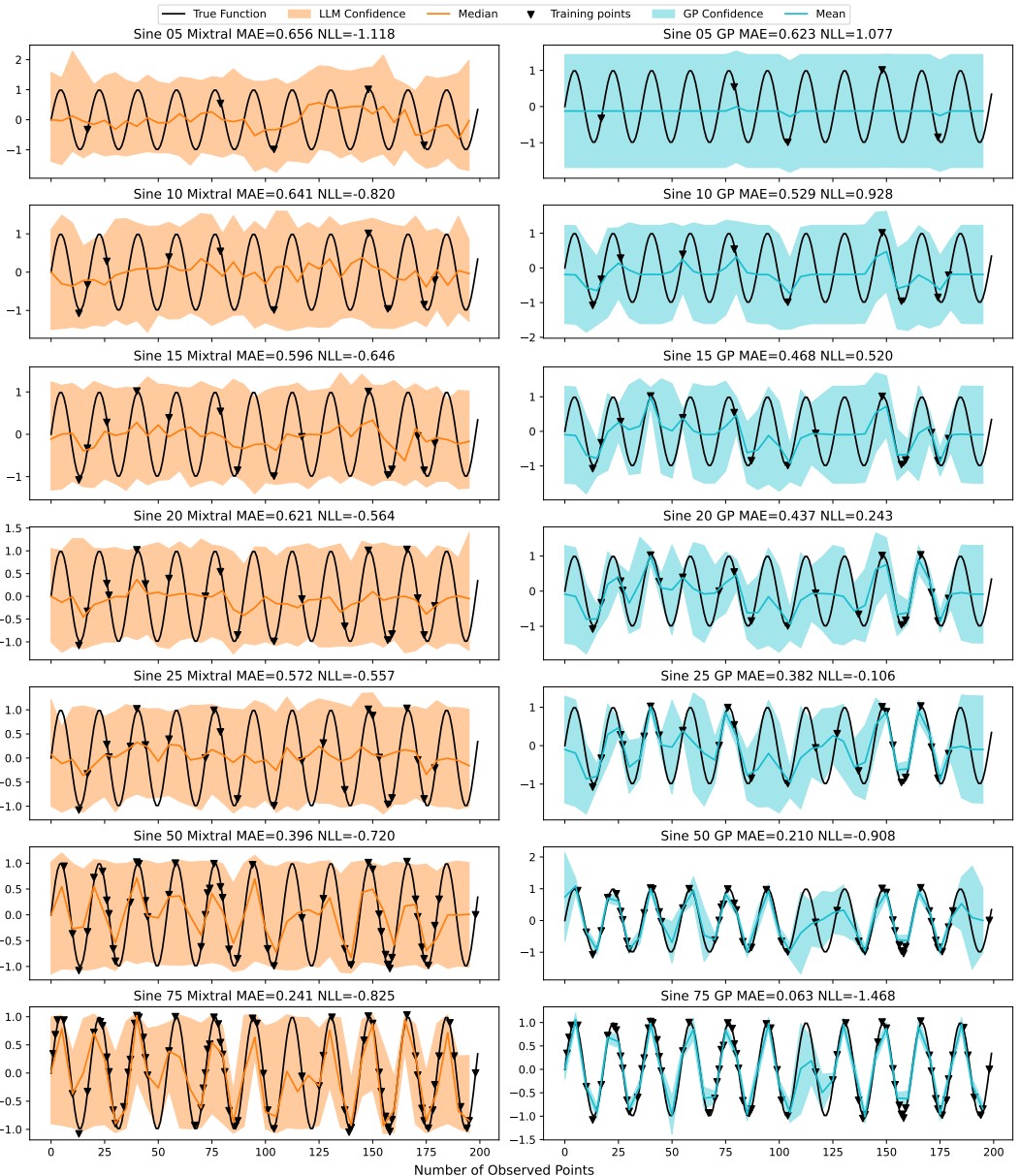

Figure H.23: MAE (lower is better) and NLL (lower is better) for the Mixtral-8×7B LLM versus a GP as a function of the number of observed points for the Sine function. The GP uses an RBF kernel with optimized length scale and noise.

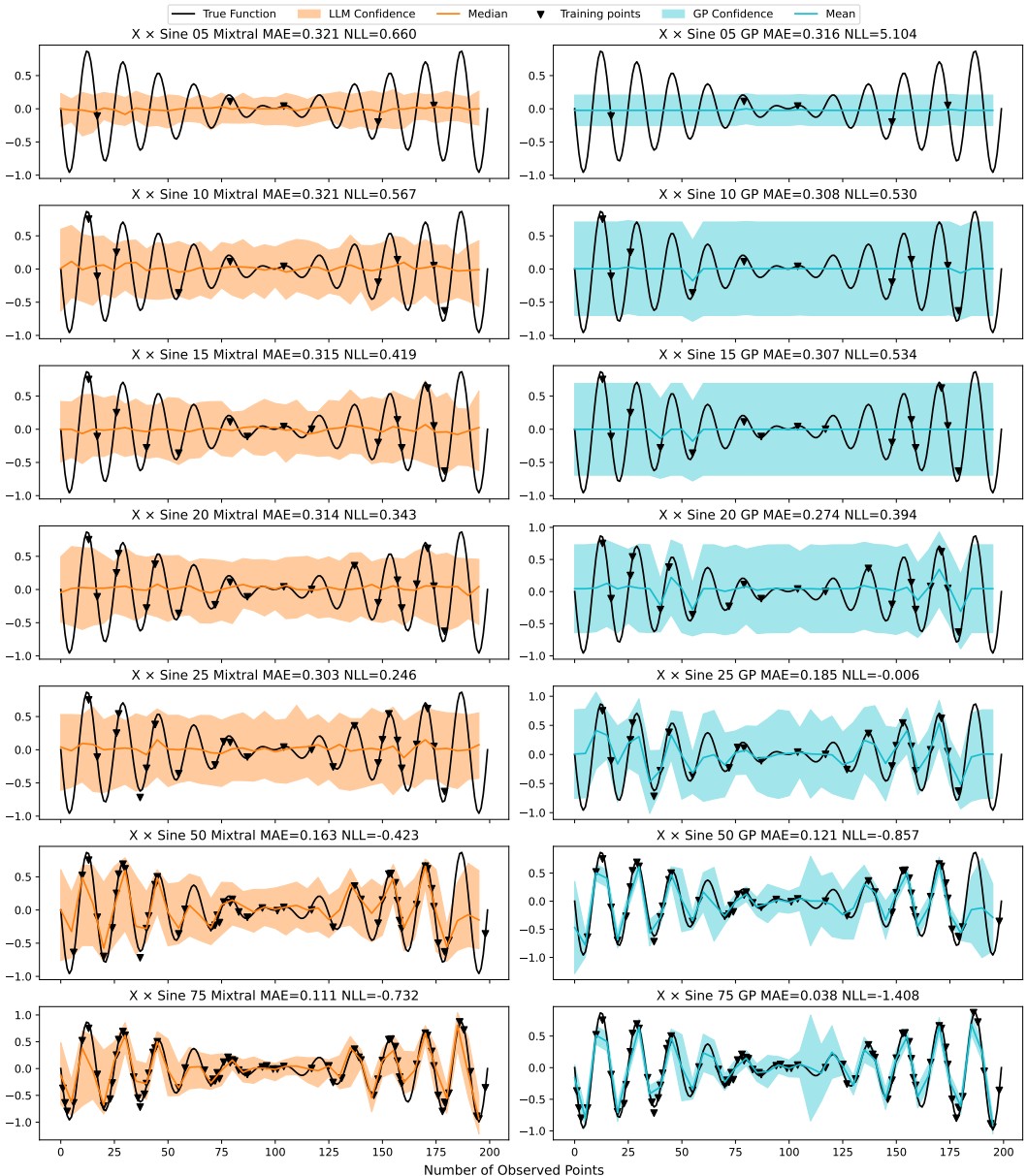

Figure H.24: MAE (lower is better) and NLL (lower is better) for the Mixtral-8×7B LLM versus a GP as a function of the number of observed points for the X × Sine function. The GP uses an RBF kernel with optimized length scale and noise.

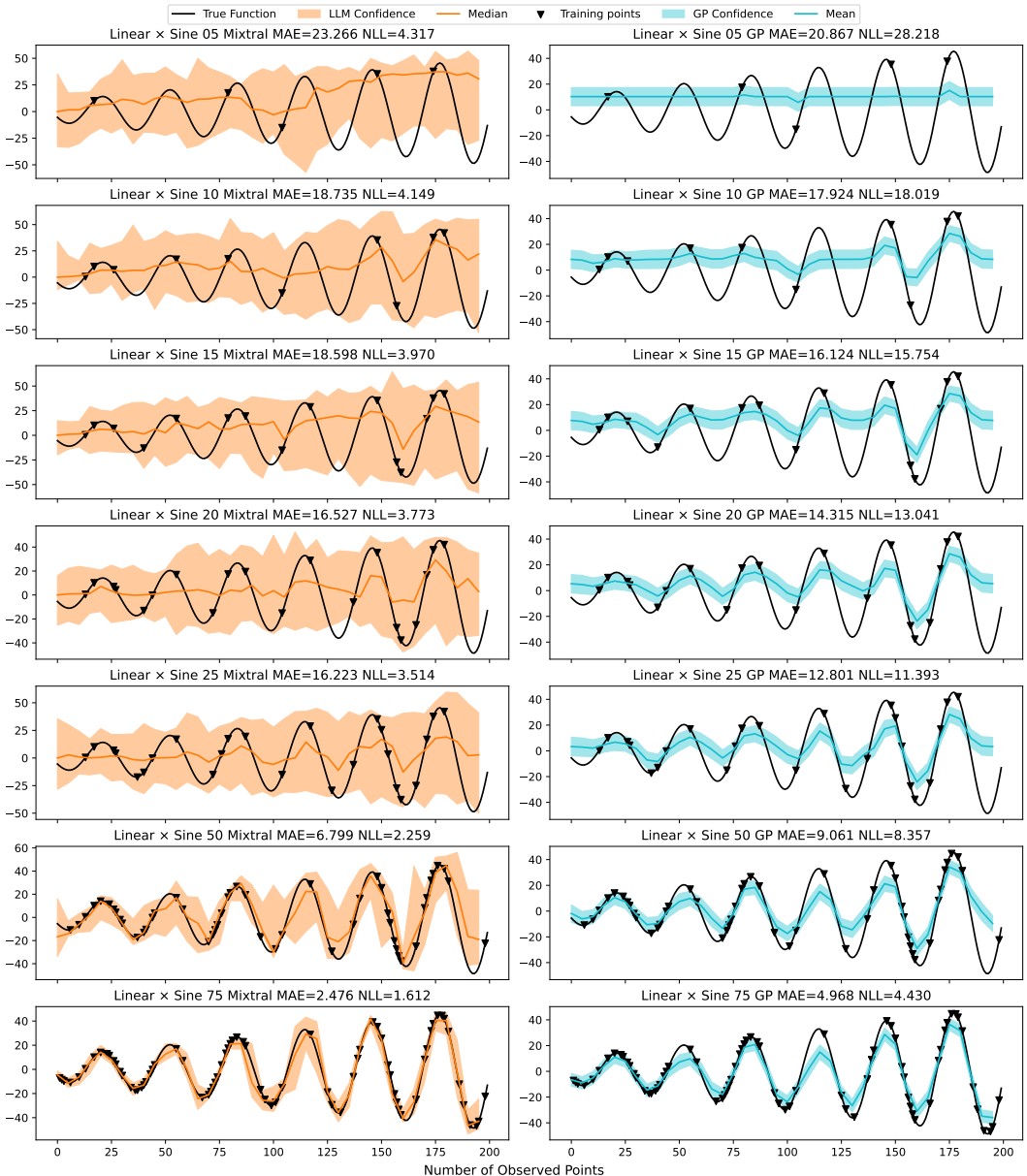

Figure H.25: MAE (lower is better) and NLL (lower is better) for the Mixtral-8×7B LLM versus a GP as a function of the number of observed points for the Linear × Sine function. The GP uses an RBF kernel with optimized length scale and noise.

Figure H.26 shows plot of NLL and MAE for the Mixtral-8×7B LLM and the RBF kernel GP for 12 for the 12 different synthetic functions.

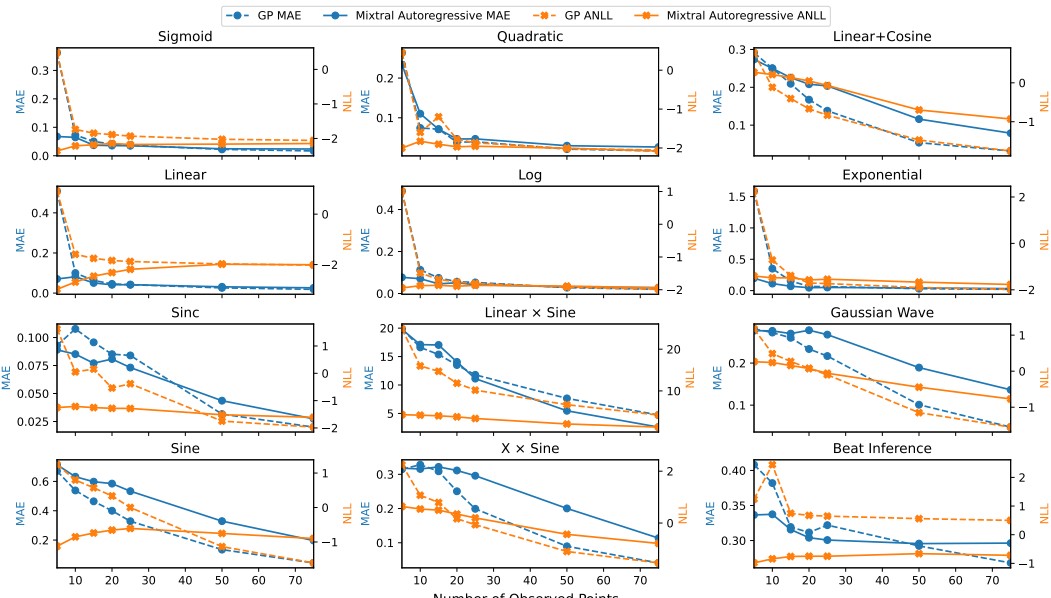

Figure H.26: MAE (lower is better) and NLL (lower is better) for the Mixtral-8×7B LLM versus a GP as a function of the number of observed points for 12 different synthetic functions. Results are averaged over three sets of random samples for the observed points. The GP uses an RBF kernel with optimized length scale and noise.

## H.2 Multimodal Predictive Experiment Details

To verify that LLMPs are able to produce non-Gaussian, multimodal predictive distributions we sampled training data from the following synthetic, bimodal generative distribution:

$$y = \frac{.05}{1 + \exp{-x}} + 0.02x + \epsilon_1(0.02x + 0.08) + 0.03\epsilon_2 \tag{H.1}$$

Where $\epsilon_1 \sim \text{Bernoulli}(p = 0.5)$ and $\epsilon_2 \sim N(0, 1)$. The Llama-3-70B A-LLMP predictive distribution using 100 training points is visualized in Figure 4 (*right*) and using 40 training points is visualized in Figure H.27.

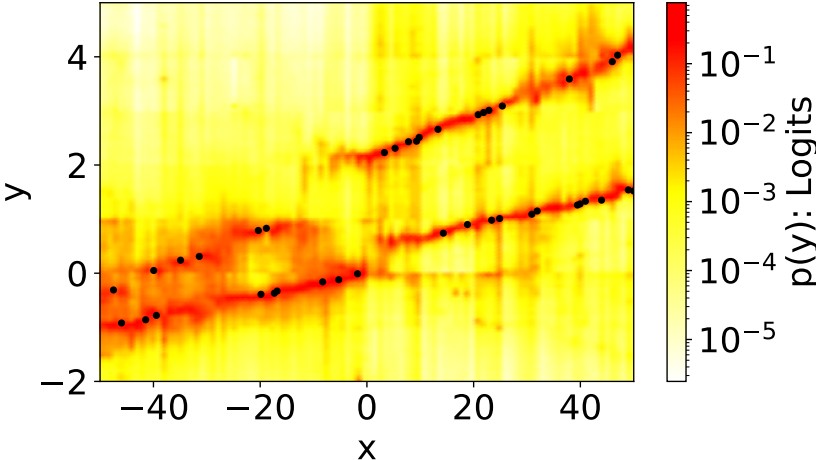

Figure H.27: Heatmap visualization of the Llama-3-70B A-LLMP predictive distribution conditioned on data from a bimodal generative process. Black dots are the 40 training points.

## H.3    Comparison to LLMTime

Figure H.28 compares A-LLMP in a temperature forecasting scenario to LLMTime. The dataset consists of 86 daily high temperature readings, obtained after the training cut-off for the Llama-2 LLM to avoid data-leakage. We use the first 50 readings for training data and ask the two methods to predict/forecast the final 36 values. The authors of LLMTime suggest the method can handle missing values by inputting NaN values in their place. Since LLMPs can work with irregularly spaced and missing data, we also compare the methods with a reduced number of irregularly spaced training points. A-LLMP wins out over LLMTime, as the log probabilities for A-LLMP are significantly better.

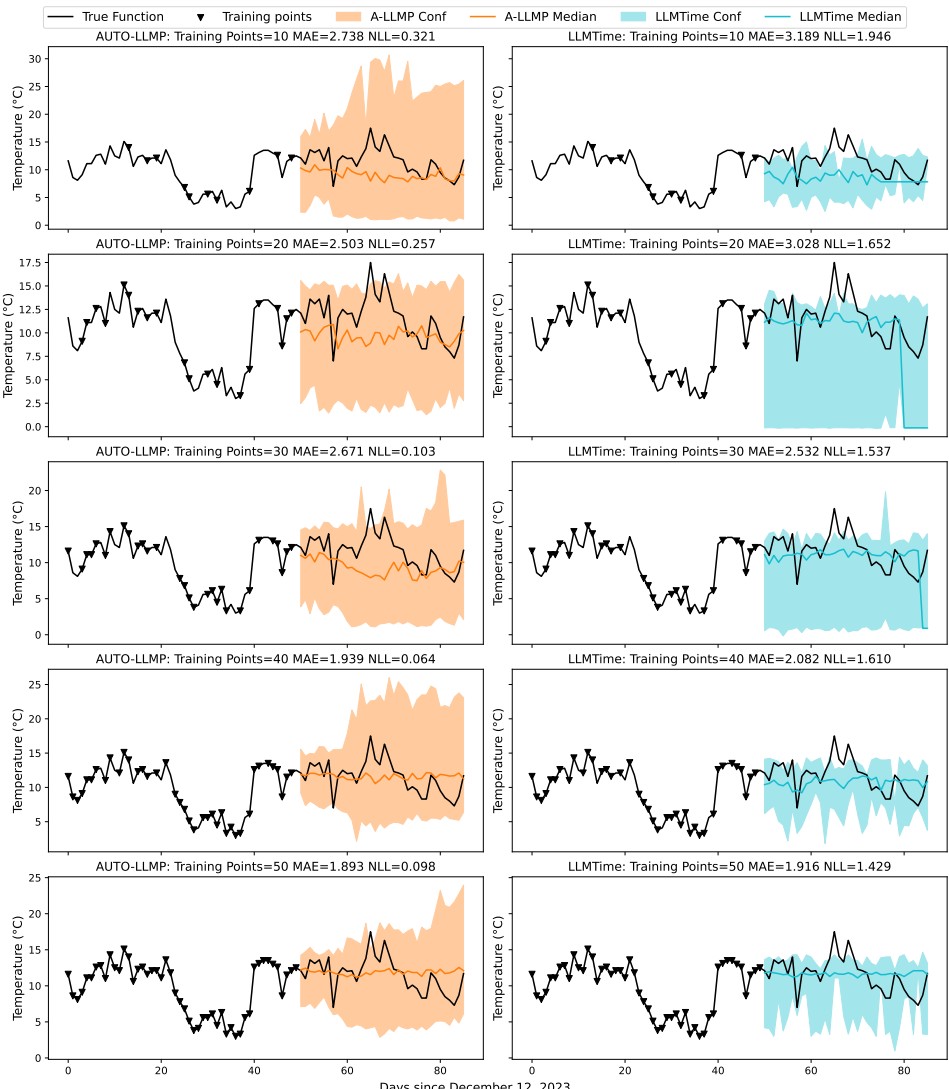

Figure H.28: MAE ↓ and NLL ↓ for A-LLMP versus a LLMTime on a dataset of daily temperatures in London, UK recorded after the release date of the LLM with a varying number of training points. The LLM is Llama-2-7B in both cases.

## H.4 Additional Image Reconstruction Results and Details

Figure H.29 depicts six image reconstruction results, all drawn from the Fashion-MNIST dataset
[12]. The $28 \times 28$ pixel images were first scaled to $20 \times 20$, due to the context size limitations of the
open-source LLMs we used in our experiments. The pixel data was then converted into prompt data
points by forming a series of (row, column, pixel value) integer tuples. We then sampled 80 pixel
locations (20%) and 200 pixel locations (50%) as observed points for the reconstruction. Each pixel
location (400 in all) was used as a target point location for independent marginal sampling with the
Mixtral-8×7B LLM.

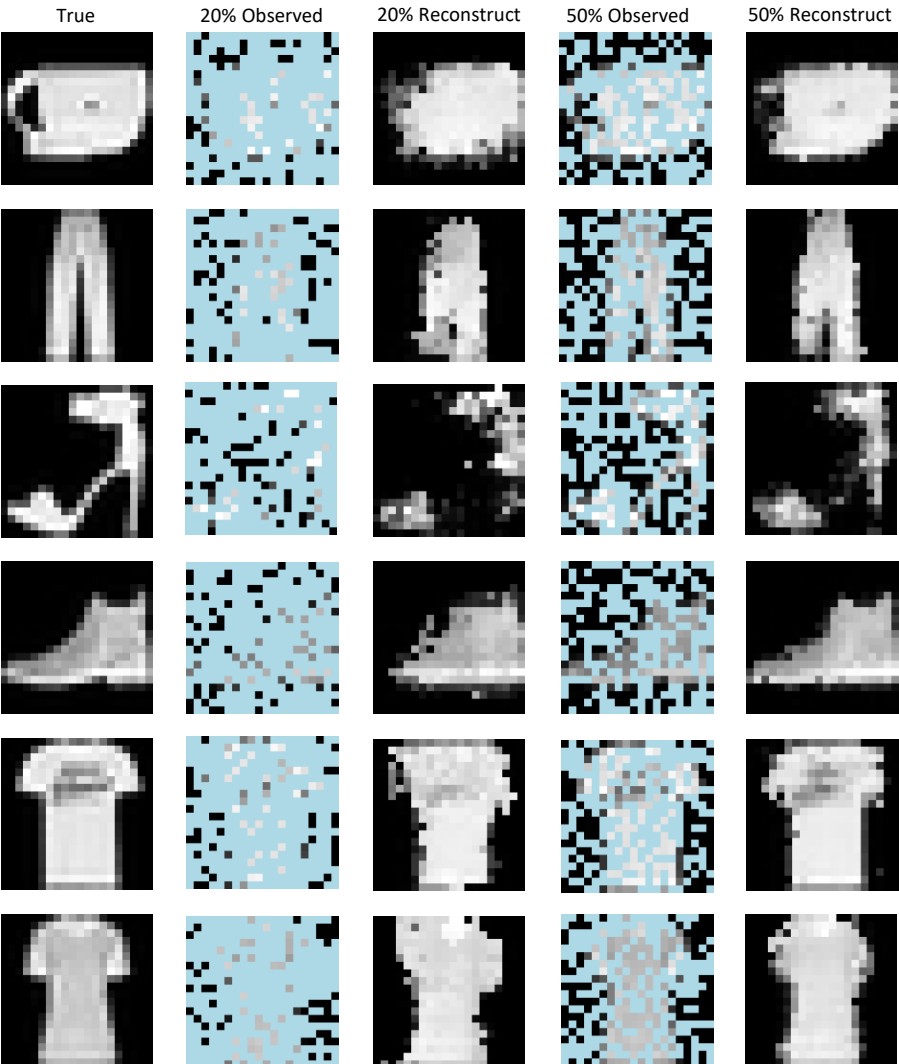

Figure H.29: Image reconstruction results for six images drawn from the Fashion-MNIST dataset
[12]. 1st column: True images. The 2nd and 4th columns are the observed pixels for the regression
task and are sampled at 20% and 50% from the true image pixels. The blue pixels indicate unobserved.
The 3rd and 5th columns show the reconstructions using the Mixtral-8×7B LLM.

## H.5 Black-box Optimization Results and Implementation Details

Black box optimization involves minimizing or maximizing a function where there is only access to the output of a function for a specified input. It is often used to optimize functions that are expensive to evaluate and the goal is to find the minimum or maximum value with the fewest number of calls to the function (often referred to as trials). To acquire the location of the next point to observe, we sample the LLM using Thompson sampling [38, 39]. Details are in Algorithm 3. We benchmark the ability of an LLM to perform black box maximization on six commonly used functions implemented in [40], including Gramacy [41], Branin, Bohachevsky, Goldstein, and Hartmann3. We compare our results using Llama-2-7B to Optuna [3], a commercial hyperparameter optimization framework. We run both methods for 100 trials and record the trial at which the the best approximation to the maximum occurs. The results are shown in Table H.8. In all cases, we obtain as good or better approximation to the true maximum value in a fewer number of trials. Note that Optuna will perform 100 trials in a few seconds while the LLM approach can take up to 2 Nvidia A100 GPU hours. However, the results show that the log likelihood of LLMPs is capable of accurately portraying regression uncertainty.

Table H.8: Black box optimization results. The number in the *Function* column indicates the number of $x$ dimensions. The *Trial* column indicates the trial at which the *Best* estimate of the maximum for each method occurred.

| Function | TRUE | Optuna | | Llama-7B | |
|---|---|---|---|---|---|
| | | Trial | Best | Trial | Best |
| Sinusoidal (1) | 1.879 | 70 | 1.879 | 23 | 1.879 |
| Gramacy (1) | 0.869 | 48 | 0.869 | 29 | 0.869 |
| Branin (2) | -0.040 | 85 | -0.041 | 70 | -0.040 |
| Bohachevsky (2) | 0.000 | 82 | -5.539 | 49 | -1.305 |
| Goldstein (2) | -3.000 | 35 | -4.876 | 31 | -3.101 |
| Hartmann (3) | 3.863 | 86 | 3.745 | 53 | 3.863 |

**Algorithm 3** Pseudocode for LLM black-box function optimization

---

**Require:** $f(\mathbf{x})$: Function to be maximized
**Require:** $\mathbf{x}_{min}$: Minimum bound on $\mathbf{x}$
**Require:** $\mathbf{x}_{max}$: Maximum bound on $\mathbf{x}$
**Require:** $T$: Number of trials (default 100)
**Require:** $M$: Number of target points (default 500)
**Require:** $C$: Number of cold start points (default 7)
   observed$_x \leftarrow$ [ ]                                           ▷ List of observed $\mathbf{x}$ values
   observed$_y \leftarrow$ [ ]                                           ▷ List of observed $y$ points
   **for** trial $\leftarrow 1$ to $C$ **do**
      $\mathbf{x} \leftarrow\sim \mathcal{U}(\mathbf{x}_{min}, \mathbf{x}_{max})$
      observed$_x$.append($\mathbf{x}$)
      observed$_y$.append($f(\mathbf{x})$)
   **end for**
   **for** trial $\leftarrow C + 1$ to $T$ **do**
      targets $\leftarrow$ [ ]                                   ▷ List of target $\mathbf{x}$ points
      samples $\leftarrow$ [ ]                            ▷ List of samples at target points
      **for** i $\leftarrow 1$ to $M$ **do**
         target$_x \leftarrow\sim \mathcal{U}(\mathbf{x}_{min}, \mathbf{x}_{max})$
         targets.append(target$_x$)
         prompt $\leftarrow$ `construct_prompt`(observed$_x$, observed$_y$, target$_x$) ▷ construct a text prompt
         samples $\leftarrow$ `Algorithm 1`($N = 1$)    ▷ Use Algorithm 1 to obtain a single sample at the
target point
      **end for**
      new_observed$_x \leftarrow$ targets[`argmax`(samples)]               ▷ Thompson sampling
      observed$_x$.append(new_observed$_x$)
      observed$_y$.append($f$(new_observed$_x$))
   **end for**
   max$_y \leftarrow$ `max`(observed$_y$)                    ▷ Best estimate of maximum value of $f$
   max$_x \leftarrow$ observed$_x$[`argmax`(observed$_y$)]     ▷ value of $\mathbf{x}$ where best estimate of maximum of $f$
occurs

---

## H.6 In-context Experiment Details and Additional Plots

For the in-context learning experiment in Section 4 we investigate LLMPs' ability to learn from similar examples in-context to predict average monthly precipitation across 13 Canadian locations [13], one from each province and territory: Alert, NU, Charlottetown, PE, Comox, BC, Goose, NL, Greenwood, NS, Keylake, SK, Montreal, QC, Ottawa, ON. Ranfurly, AB, Saint John, NB, Thompson, MB, Whitehorse, YK, and Yellowknife, NT. For each location, we use the Mixtral-8×7B A-LLMP to forecast 32 months of average precipitation values given the previous four month observations taken from a random historical three-year period between 1913-2017 (conditional on data availability). It is then provided with 1-12 examples of random three year periods of historical values from the same location in-context. An example prompts for 0, 1 (1976-1978) and 2 (1976-1978, 1949-1951) examples are:

1. "Monthly total of daily adjusted rainfall, mm. \n1976-1978:\n",

2. "Monthly total of daily adjusted rainfall, mm. \n1967-1969:\n0,0.3\n1,0.6\n2,1.3 \n 3,0.6\n4,31.7\n5,59.9\n6,135.4\n7,107.7\n8,78.3\n9,40.7 \n10,37.3\n11,5.4\n12,1.0 \n 13,41.4\n14,0.3\n15,29.2\n16,41.3\n17,67.8\n18,137.8\n19,139.9\n20,91.4\n21,143.1\n22,18.8 \n23,0.9\n24,0.6\n25,14.0\n26,4.0\n27,6.2\n28,45.1\n29,98.3\n30,97.0\n31,160.4\n32,116.3\n 33,22.4\n34,51.8\n35,38.1\n1976-1978:\n",

3. "Monthly total of daily adjusted rainfall, mm. \n1967-1969:\n0,0.3\n1,0.6\n2,1.3\n 3,0.6\n4,31.7\n5,59.9\n6,135.4\n7,107.7\n8,78.3\n9,40.7\n10,37.3\n11,5.4\n12,1.0\n 13,41.4\n14,0.3\n15,29.2\n16,41.3\n17,67.8\n18,137.8\n19,139.9\n20,91.4\n21,143.1\n22,18.8\n 23,0.9\n24,0.6\n25,14.0\n26,4.0\n27,6.2\n28,45.1\n29,98.3\n30,97.0\n31,160.4\n32,116.3\n 33,22.4\n34,51.8\n35,38.1\n 1949-1951:\n0,1.6\n1,0.0\n2,2.5\n3,2.1\n4,22.0\n5,51.7\n6,83.4\n7,113.3\n8,75.5\n9,34.7\n10,4.7\n 11,1.4\n12,1.1\n13,0.0\n14,0.8\n15,9.5\n16,33.3\n17,92.6\n18,118.5\n19,70.3\n20,34.6\n21,58.2\n 22,62.4\n23,8.5\n24,0.3\n25,7.4\n26,8.0\n27,30.6\n28,49.3\n29,40.0\n30,82.5\n31,97.1\n32,71.5\n 33,17.1\n34,32.1\n35,10.1\n1976-1978:\n".

Results are presented in Figure 8, Figure H.30 and Figure H.31.

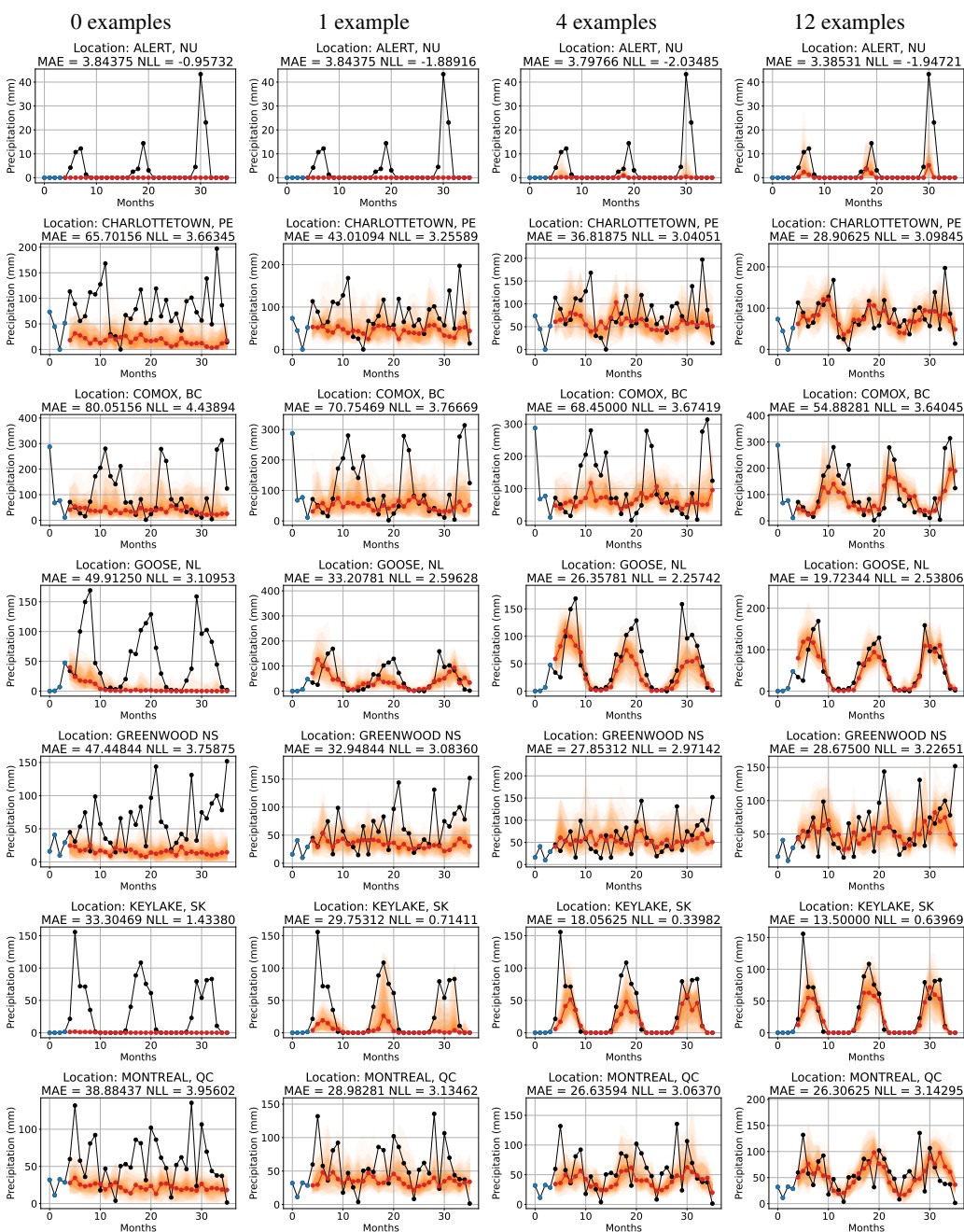

Figure H.30: Visualizations of the predictions given by the Mixtral-8×7B LLMP for seven locations locations accross Canada. Blue and black circles are training and test points, respectively. Red circles are median predictions and shaded areas indicate tenth-percentiles over 30 samples.

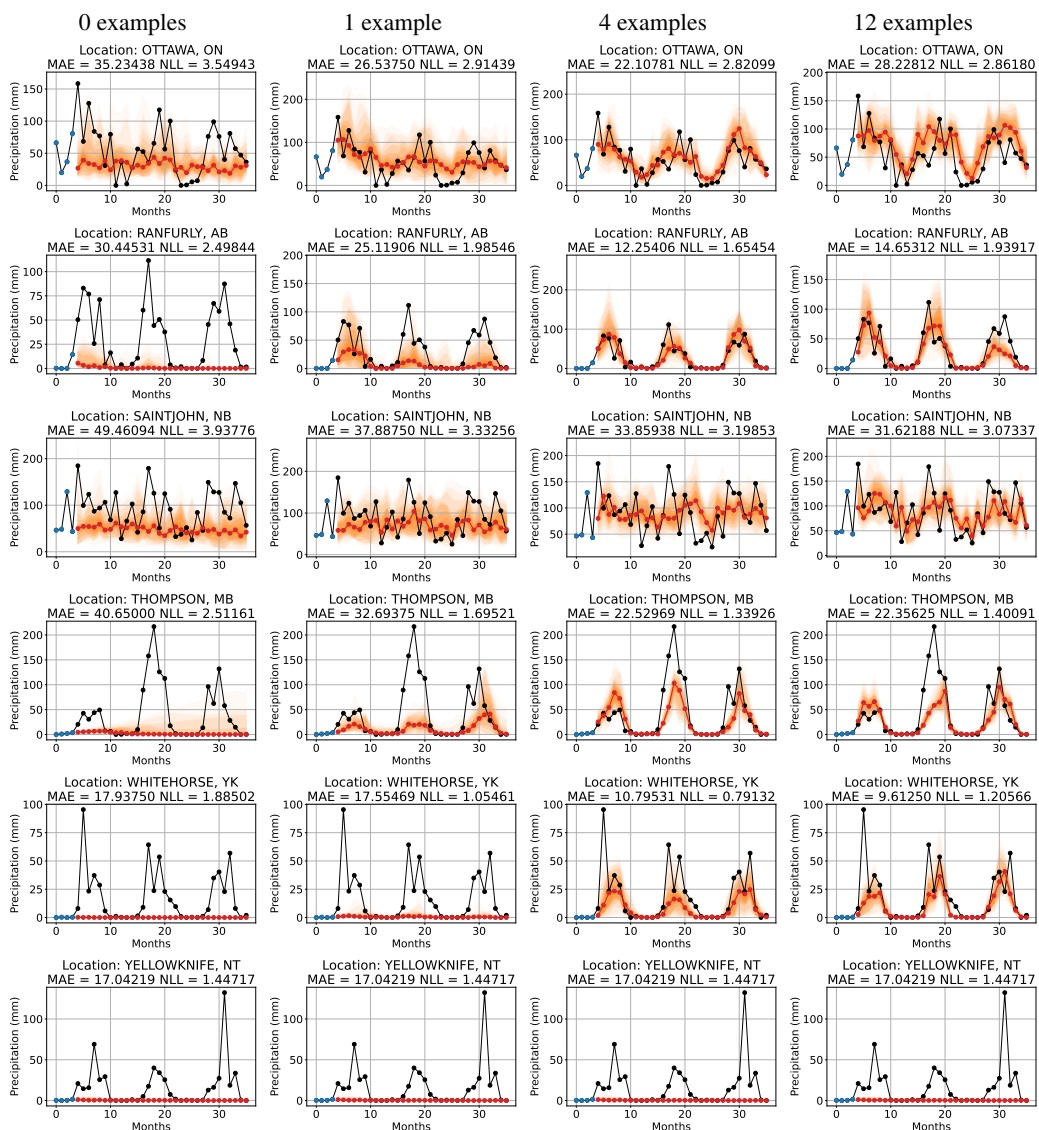

Figure H.31: Visualizations of the predictions given by the Mixtral-8×7B LLMP for six locations locations accross Canada. Blue and black circles are training and test points, respectively. Red circles are median predictions and shaded areas indicate tenth-percentiles over 30 samples.

# I Conditioning on Text Details and Additional Experiments

## I.1 Scenario-conditional Predictions Details and Additional Experiments

For the scenario-conditional predictions experiment in Section 5, we examine the influence of text providing information about various synthetic problem settings on the predictive distribution of an Llama-3-70B LLMP. In all of the following examples, we provide the same two synthetic training points, $(1, 2.53)$ and $(2, 2.21)$ to the LLM Process but change the prompting text that comes before the training data. We then use A-LLMP to forecast trajectories integer 50 steps ahead. Prompts were prepended to the standard data formatting scheme used for LLMPs (see Appendix C).

The prompts provided to the LLMP visualized in Figure 9 are:

1. "" (i.e. no text);

2. 'The following are daily temperature measurements from Montreal in January in degrees Celsius"

3. "The following are daily temperature measurements from Montreal in May in degrees Celsius"

4. "In the following series, the first number is the number of Months from January and the second is the Monthly precipitation measurements in inches from San Diego, CA"

5. "In the following series, the first number is the number of Months from February and the second is the Monthly precipitation measurements in inches from Singapore"

The prompts visualized in Figure 1 are:

1. "The following are daily stock prices from a financial time series"

2. "The following are daily stock prices from a financial time series for a company that eventually goes out of business"

3. "The following are daily average stock prices from a financial time series for a company whose stock price goes to zero on day 30"

**Lynx Hare Population Forecasting:**  Similar to the previous experiment, this experiment examines to what extent the predictive posterior of an LLM Process is influenced by textual information about the problem provided in the prompt. We preface the prompt with three different strings:

1. "" (i.e. no text);

2. "The following are samples from lynx-hare populations"

3. ''The following are samples from the famous Canadian Hudson Bay Lynx-Hare population dataset. When hare increases, lynx increases. The first number of two is the year. The second number is the lynx population. It follows the pattern when lynx population increases, hare decreases"

Figure I.32 shows the predictive distribution of the LLM with 10 and 50 observed points. As the specificity of the text increases from L to R, the posterior entropy decreases, and structure of the samples changes dramatically.

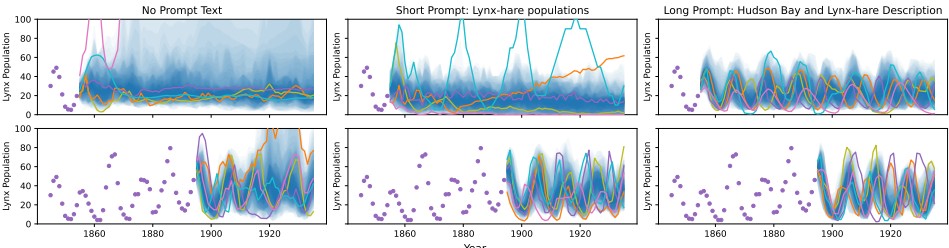

Figure I.32: Results of condition on both text and numerical data simultaneously, on the Mixtral model. Observed points are in purple. Colored lines show sampled trajectories. The blue shading is a visualization of percentiles based on 50 samples. *Top:* Conditioning on 10 observed points. *Bottom:* Conditioning on 50 observed points. The predictive distribution changes as more information about the problem is added to the prompt.

### I.2 Labelling Features Using Text Details and Additional Plots

In the experiments in section Section 5 we examine the performance of a Mixtral-8x7B Instruct I-LLMP on predicting American housing prices. The dataset [16] contains 39980 housing prices and various variables around housing and demographics for the top 50 American cities by population. This dataset was generated on 12/09/2023, however it contains data from the 2020 US Census and the 2022 American Community Survey (ACS). It is possible that data within this dataset was used to train Mixtral-8x7B but it is very unlikely that it was trained on the exact strings presented in this experiment.

For each prediction task, we show the I-LLMP 10 randomly selected training examples from the dataset and predict on 20 randomly selected test examples. In the prompt, before the numerical value (price) we provide a string which encodes the datapoint index/features that the model can use. For our first experiment we examine the behaviour of the LLMP when more features are added to the prompt. We experiment with five ways of indexing the training and test points illustrated by the following training examples;

1. "32.74831, -97.21828, Price: 224900.00"
2. "Location: Fort Worth, Texas, Latitude: 32.74831, Longitude: -97.21828, Price: 224900.00"
3. "Location: Fort Worth, Texas, Latitude: 32.74831, Longitude: -97.21828, Zip Code: 76112, Median Household Income: 71452.0, Price: 224900.00"
4. "Location: Fort Worth, Texas, Latitude: 32.74831, Longitude: -97.21828, Zip Code: 76112, Median Household Income: 71452.0, Zip Code Population: 42404 people, Zip Code Density: 1445.0 people per square mile, Price: 224900.00"
5. "Location: Fort Worth, Texas, Latitude: 32.74831, Longitude: -97.21828, Zip Code: 76112, Median Household Income: 71452.0, Zip Code Population: 42404 people, Zip Code Density: 1445.0 people per square mile, Living Space: 1620 square feet, Number of Bedrooms: 3, Number of Bathrooms: 2, Price: 224900.00"

This procedure is repeated 10 times to compute statistics. Results from this experiment are presented in Figure 10 (*left, centre*) and in I.34. We also ran this experiment using Mixtral-8x7B and found that the performance, shown in Figure I.33, was not as good as with the instruction tuned version of Mixtral-8×7B.

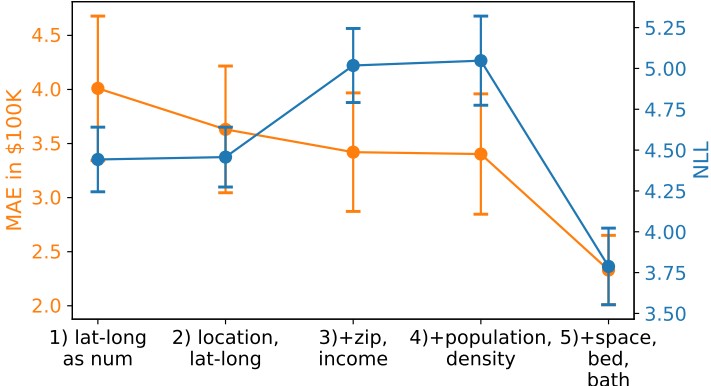

Figure I.33: Average MAE and NLL performance of the Mixtral-8x7BLLMP over 10 experiments with error bars representing the standard error.

An additional experiment is presented in Section 5 to see examine the effect of adding text labels to the features. This experiment was run on 10 new random datasets providing the LLMP with either labeled or unlabelled numerical features. Due to the results of the previous experiment, a Mixtral-8x7B Instruct LLMP was used for this experiment. The following are example training strings for the four cases examined:

a. "30.45738, -97.75516, Price: 385000.00"

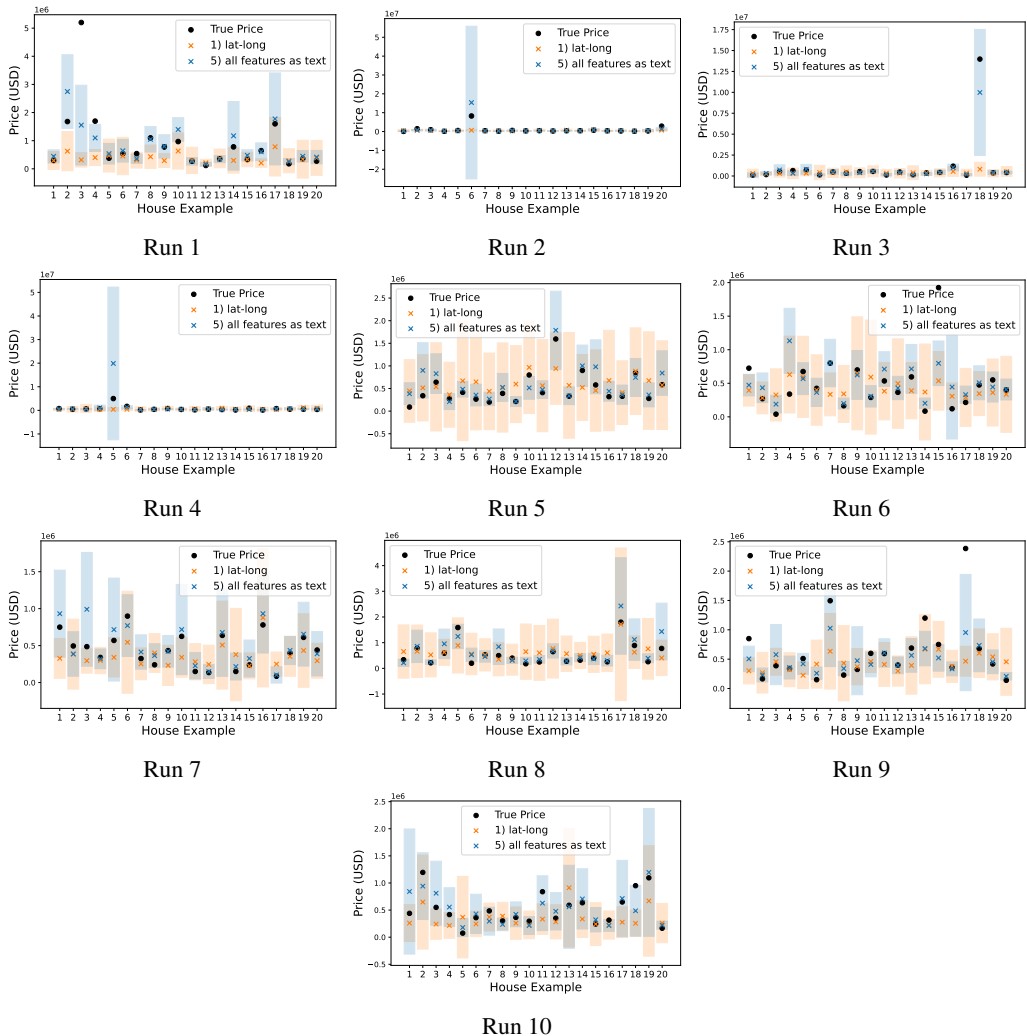

Figure I.34: Results of 10 runs using Mixtral-8x7B Instruct I-LLMP predicting US housing prices for 20 random houses from [16]. Predictions are visualized using index style 1) and 5). Xs are mean predictions using 30 samples from the LLMP and error bars indicate 2 standard deviations.

b. "Location: Austin, Texas, Latitude: 30.45738, Longitude: -97.75516, Price: 385000.00"

c. "30.45738, -97.75516, 78729, 107830.0, 30907, 1216.1, 1349, 3, 2, Price: 385000.00"

d. "Location: Austin, Texas, Latitude: 30.45738, Longitude: -97.75516, Zip Code: 78729, Median Household Income: 107830.0, Zip Code Population: 30907 people, Zip Code Density: 1216.1 people per square mile, Living Space: 1349 square feet, Number of Bedrooms: 3, Number of Bathrooms: 2, Price: 385000.00".

Results of this experiment are presented in Figure 10 (*right*).

## J  Additional Comments on Limitations and Societal Impact

**Limitations** As mentioned in the main text along with the flexibility of LLMs, LLMPs inherit their drawbacks. An additional drawback of using LLMs for probabilistic regression is that results from LLMPs are inherently less interpretable than from methods like Gaussian processes where we explicitly encode priors. As with other black-box methods, we must, at the moment, rely on demonstrating empirically that it makes well-calibrated predictions.

**Societal Impact** Our work has demonstrated a new and useful zero-shot approach for generating probabilistic predictions using plain language to augment numerical data. It has the potential to allow practitioners from fields such as medical research and climate modelling to more easily access probabilistic modelling and machine learning. We hope that such an impact would help researchers improve the lives of all humans by tackling the problems that humanity faces today.

Like all machine learning technology, there is potential for abuse, and possible consequences from incorrect predictions made with LLMPs. Due to the black-box nature of the method, we do not know the biases in the underlying LLMs used and what effect they may have on LLMPs output. However, LLM researchers are striving to make LLMs more fair and equitable. An open area of research is whether LLM biases propagate to LLMP predictions and whether de-biasing LLMs helps to fix such an issue.

