# OpenReview forum: "LLM Processes: Numerical Predictive Distributions Conditioned on Natural Language"
_NeurIPS.cc/2024/Conference — NeurIPS 2024 poster_

### Official Review · Reviewer_mWHr · 2024-07-11

**Soundness:** 4
**Presentation:** 3
**Contribution:** 3
**Rating:** 6
**Confidence:** 3

**Summary:**

This paper focuses on the setting where the goal is to use LLMs to make numerical predictions, as these models can be grounded by the provision of side-information (given in text) as well as other information that may be learned during their pretraining.

They propose LLM Processes, an approach to apply LLMs to numerical tasks, such as density estimation and multivariate time series tasks. In computing their densities over continuous valued numerical inputs, they employ binning as is done in LLMTime, although they do not rescale to remove decimal points. They also add an additional terminal special token (<t>).
In selecting their input prompt formats, they use separators between each x, y and (x,y) pair, order points by distance to the current point, scale the y values to be closer to [0, 1] and not incorporate -1 values.

On many 1D synthetic tasks, LLMP demonstrates matching or better performance when compared to GPs with a RBF kernel. Their approach outperforms the most related baseline of LLMTime in terms of both NLL and MAE in predictions on a Weather time series benchmark. Their experiments also outperform a GP (with RBF kernel) on a multidimensional input/output task of simultaneous temperature, rainfall and wind speed regression task.

They also provide experiments looking at the ability of LLMP to handle textual information (alongside numerical inputs); they provide descriptions of the numerical features, which cannot be used in baseline forecasting approaches that do not understand text.

**Strengths:**

1. Good empirical results showing that the proposed approach improves over the baselines of LLMTime and a BP with an RBF kernel across a variety of univariate and multivariate forecasting tasks.
2. Ablations supporting particular choices for input formatting and scaling

**Weaknesses:**

Slightly unclear in the evaluation; please see questions below.

**Questions:**

1. It’s not immediately clear to me what the takeaways from Figure 9 are. What are the axes for each of the subplots?
2. My current understanding is that incorporating some additional textual information should meaningfully change the predictive distribution, which seems to occur. Is there any way to measure how correct this change in distribution is?
3. For instance, in Figure 9d, it seems that the median from 5 random samples seems to be roughly constant (around 2.5-3), which does not seem to match the ground truth values in 9f. Is this case just a failure of the LLM Process?

**Limitations:**

Yes, limitations are adequately addressed.

---

> ### Author Rebuttal · Authors · 2024-08-02
>
> Thanks for taking the time to review our paper and for the positive words about our contributions. We address your questions below.
>
> Figure 9 is intended to demonstrate how predictions from an LLMP can be influenced by conditioning on various scenarios communicated via text prompts to the model. This experiment is a qualitative investigation with no specific ground truth information. However, we hope to show that a prompt containing two synthetic initial data points plus some descriptive text will yield samples that are meaningful.
>
> We provide units for the axes in the prompts (detailed in Appendix J.1) as follows:
> - Figure 9b) “The following are daily temperature measurements from Montreal in January in degrees Celsius”
> - Figure 9c) “The following are daily temperature measurements from Montreal in May in degrees Celsius”
> - Figure 9d) “In the following series, the first number is the number of Months from January and the second is the Monthly precipitation measurements in inches from San Diego, CA”
> - Figure 9e) “In the following series, the first number is the number of Months from February and the second is the Monthly precipitation measurements in inches from Singapore”
>
> We will add these units to the plots in the next revision.
>
> You are correct that the median response in Figure 9d) does not match the ground truth in Figure 9f) particularly well. They match in the sense that both figures indicate that the precipitation is quite low all year round in Singapore, but it does not match the minor rising and falling trend. We think that this can be attributed to the fact that since the synthetic datapoints also do not match actual values very well, the LLMP had difficulty reconciling this mismatch between values and text information. We included this experiment primarily to demonstrate that the model is aware of this seasonal trend in San Diego vs. Singapore (Singapore has one of the least seasonal trends in precipitation on earth) and that can reasonably modify the LLMPs predictive distribution on the same collection of data values by varying the prompts.
>
> Different from Figure 9, Figure 10 is a quantitative investigation on how incorporating textual information can improve predictive performance. In this experiment, we measure how correct this change in predictive distribution is by examining improvements in the Negative Log-likelihood (NLL) and the Mean Absolute Error (MAE) given by the predictive distribution when text is included in the prompt versus when it is excluded. Figure10 shows that both these metrics improve (i.e. are lower) when conditioning on text.
>
> Please let us know if you have any more comments or questions. If we have adequately addressed your concerns, we kindly ask that you consider revising your score.

---

> > ### Comment · Reviewer_mWHr · 2024-08-08
> > **Reviewer Response**
> >
> > Thank you for the clarifications about the evaluation and axes meaning in the figures. I'll maintain my stance of a weak accept; I think there are useful comparisons to prior work and ablations that illuminate some of both the abilities and failures of the application of LLMs to numerical prediction.

---

### Official Review · Reviewer_Qkoz · 2024-07-12

**Soundness:** 4
**Presentation:** 4
**Contribution:** 4
**Rating:** 9
**Confidence:** 4

**Summary:**

This paper proposes using LLMs to model joint distributions over numerical outputs while conditioning on potentially multiple covariates per data point. This is achieved through tokenizing a series of input and output pairs and then decoding corresponding outputs for another input. Outputs for numerical values are achieved by treating the number as a sequence of digits, each of which are represented with a token. The method is empirically validated with additional experiments showcasing capabilities of conditioning on additional contextual information, such as text describing the process.

**Strengths:**

The problem statement is innovative and clearly investigates many different ablations and alternatives. Extensive empirical results are shown, investigating different failure modes and comparing performance to traditional methods. The method demonstrates great versatility and interesting applications. Additionally, the paper itself is very well-written and clear in its presentation.

**Weaknesses:**

The main weakness in the approach that I potentially see is the runtime. From what I could tell, there is no runtime results in either the main paper or the appendix. For processing a sequence of input-output pairs, what is the general runtime exhibited, and how does it change as the sequence grows?

**Questions:**

See weaknesses.

**Limitations:**

The authors extensively discussed the limitations and societal impact of the model.

---

> ### Author Rebuttal · Authors · 2024-08-02
>
> Thanks for taking the time to review our paper and for the positive words about our contributions. We address your questions below.
>
> **The main weakness in the approach that I potentially see is the runtime. From what I could tell, there is no runtime results in either the main paper or the appendix. For processing a sequence of input-output pairs, what is the general runtime exhibited, and how does it change as the sequence grows?**
>
> We agree that this is the main weakness our method. We view the main use case of LLMPs as being when the benefits of incorporating textual information into your regression problem outweigh the significant computational expenses involved. Appendix G “Additional Implementation Details” does contain some example runtimes. It states that processing times vary as a function of the:
> - GPU used.
> - length of the prompt.
> - number of target points queried.
> - number of tokens required to be generated for a particular target point.
> - the number of samples taken at each target point
> - and whether independent or autoregressive sampling is used.
>
> However, Appendix G does not specifically address your questions, so below is some new data that hopefully will give you a more tangible understanding of runtime performance. Note that this new data is quantitative as oppose to general "Big O notation".
>
> The table below shows the time to load the LLM into GPU memory, the time for the LLM to generate all samples at all target points, and the time to compute the probability distribution over the true target points. All runs used the Llama-2-7B LLM and were executed on an NVIDIA 3090 GPU with 24GB of memory with a batch size of 10. All times are in seconds.
>
> | Function                                         | Model  | Load (s) | Sample (s) | Likelihood (s) |
> |:--------------------------------------------------|:--------:|:----------:|:------------:|:------------------------:|
> | Quadratic - 10 Training Points, 40 Target Points         | I-LLMP | 5        | 81         | 1                      |
> | Quadratic - 10 Training Points, 40 Target Points         | A-LLMP | 5        | 170        | 3                      |
> | Quadratic - 50 Training Points, 40 Target Points         | I-LLMP | 5        | 259        | 4                      |
> | Quadratic - 50 Training Points, 40 Target Points         | A-LLMP | 5        | 354        | 7                      |
>
> From the table, we can see that the longer the prompt, the longer the computation time for each target point. For independent sampling (I-LLMP), the prompt length is constant and is only a function of the number of training points as each target point is processed independently.  For autoregressive sampling (A-LLMP), the prompt length is a function of both the number of training points and the number of target points since each target point is appended to the prompt as it is sampled.
>
> We will add these details in the next revision.

---

> ### Comment · Reviewer_Qkoz · 2024-08-10
>
> Thank you for the response to my question, this does indeed directly communicate the type of information I wanted to see regarding this issue.
>
> As far as the paper, I do still feel strongly that this is innovative work with a great deal of experimental results backing the empirical findings. As such, I maintain my original score.

---

### Official Review · Reviewer_dL9w · 2024-07-27

**Soundness:** 4
**Presentation:** 3
**Contribution:** 3
**Rating:** 7
**Confidence:** 3

**Summary:**

This paper investigates the regression problem in large language models via in-context learning. They evaluate a variety of regression tasks such as for-casting and time series prediction, multi-dimensional regression, and more. They look into prompt engineering exploiting both numerical examples and their textual explanation for eliciting coherent predictive distributions.

**Strengths:**

-The problem of looking into the regression capabilities of language models in their in-context learning is very interesting and important.
-the paper provides a large amount of experimental resutls and research work.

**Weaknesses:**

-some parts of the paper were unclear to me, especially the experimetnal part.
-Though the related works are covered well in terms of citation the actual resutls are not compared to the previous ones. see below my questions.

**Questions:**

If I correctly understand the experimetnal part reports resutls on both training and only prompting. In this case make these two paradigms more clear in the organization of the experimental results.

The paper that is cited [21] : their experiments should be comparable to this work, why the results are not compared? while you apply more technical variations here what is the main message and your new contribution compared to them? can you put their results and yours side by side and compare them?


[21] From words to numbers ....

---

> ### Author Rebuttal · Authors · 2024-08-02
>
> Thanks for taking the time to review our paper and posing insightful questions. Answers to your questions are below:
>
> **If I correctly understand the experimetnal part reports resutls on both training and only prompting. In this case make these two paradigms more clear in the organization of the experimental results.**
>
> Apologies if the organization of the experimental results section was not clear. We decided to organize our experiments into two main sections: the first section examines how LLM Processes (LLMPs) perform on purely numerical tasks, while the second section examines the influence of conditioning LLMPs on text and similar examples in-context.  Note that for our proposed methods (I-LLMP and A-LLMP), no training is ever involved. These methods use prompting only. Only the Gaussian Process hyperparameters are trained to compare with LLMPs. We will clarify this point at the start of the experimental sections in the next revision of the paper.
>
> **The paper that is cited [21] : their experiments should be comparable to this work, why the results are not compared? while you apply more technical variations here what is the main message and your new contribution compared to them? can you put their results and yours side by side and compare them?**
>
> The primary reason we did not compare to the results in the “From Words to Numbers: Your Large Language Model Is Secretly A Capable Regressor When Given In-Context Examples” paper is that it was released on arXiv on April 11, 2024 only a little more than a month before the NeurIPS submission deadline of May 22, 2024. By the time we became aware of the paper, we had finished running our experiments and had started writing up the paper.
>
> However, below are the results of a new experiment where we compare our results to theirs on the Original #1 dataset (see section 4.2 in the “From Words to Numbers” paper). The experimental set-up is as follows: There are 100 trials with each trial consisting of 50 training points and a single target point. The training and target points for each trial are randomly generated using the function described in Equation 2 in the “From Words to Numbers” paper.  We use the code from their paper to generate the data and evaluate their approach and compare it to ours using identical numerical data. We use the Llama-2-7B LLM for both methods to ensure a fair comparison.
>
> Our method (I-LLMP) achieved lower Mean Absolute Error (MAE) on 78 of the 100 trials when compared to their method. When the errors are averaged over the 100 trials, our average error was 0.836 and theirs was 3.137. These results indicate that our LLMP approach is clearly superior to the approach employed in the “From Words to Numbers” paper. This is due to the facts that a) we sort the training points according to distance to the current target point when creating the prompt whereas they do not, and b) we form a distributional estimate for the predicted point and then take the median sample value as the best estimate, whereas they generate a single point estimate. We will include this comparison in the updated version of our paper.
>
> In our paper, we also compare to the LLMTime method in Figure 5 and Gaussian Processes in Table 1 and Figure 7. These results show that we significantly outperform LLMTime, especially in terms of negative log likelihood (NLL) and are better overall when compared  to Gaussian Processes on a wide variety of functions.
>
> While we also show that LLMs are capable of non-linear regression, our contributions go significantly further than both the “From Words to Numbers” paper and LLMTime:
> - Our primary contribution is to condition on problem relevant text and demonstrate how this can improve prediction performance. Their work does not consider this.
> - Our work presents how to elicit full numerical predictive distributions from LLMs using two different methods (sampling-based and logit-based) while their method employs only point estimates for predictions.
> - We perform a comprehensive analysis of various prompt formats including ordering and scaling. They do no such experimentation and use only a single fixed prompt format.
> - We present a novel auto-regressive LLM sampling approach that yields superior results to the independent sampling approach used in their paper.
> - We compare our results to Gaussian Processes, viewed as the gold standard for probabilistic regression, while they compare to only simpler approaches.
>
> Please let us know if you have any more comments or questions. If we have adequately addressed your concerns, we kindly ask that you consider revising your score.

---

> > ### Comment · Reviewer_dL9w · 2024-08-13
> >
> > Thank you for addressing my question and adding the new resutls. I have raised my score.

---

### Author Rebuttal · Authors · 2024-08-02

We thank the reviewers for the time and effort they put into reading and commenting on our paper. Our work has presented a new zero-shot approach for generating probabilistic predictions with LLMs using plain language to augment numerical data. Reviewers believe that our work is both important and innovative, with Reviewer Qkoz stating that the “problem statement is innovative” and reviewer dL9w saying that “The problem of looking into the regression capabilities of language models in their in-context learning is very interesting and important.” All reviewers agreed that the experiments, ablation studies, and competitive performance comparisons were extensive and comprehensive. Reviewer Qkoz added that “The method demonstrates great versatility and interesting applications.”

We believe that we have addressed all of the reviewer’s questions and concerns in the rebuttal. In addition, we added two new experiments to support some of the questions raised. The first new experiment shows that our LLM regression approach is superior to the one described in the recent “From Words to Numbers: Your Large Language Model Is Secretly A Capable Regressor When Given In-Context Examples” paper, and the second new experiment provides processing times for a representative use case.

Please let us know if there are any further questions.

---

### Decision · Program_Chairs · 2024-09-25

**Decision:**

Accept (poster)

**Comment:**

The paper studies the use of LLMs and in-context learning for regression problems. The authors look into prompt engineering exploiting both numerical examples and their textual explanation for eliciting coherent predictive distributions. They evaluate a variety of regression tasks such as for-casting and time series prediction, multi-dimensional regression, and more. On many 1D synthetic tasks, LLMP demonstrates matching or better performance when compared to GPs with a RBF kernel. Their approach outperforms the most related baseline of LLMTime in terms of both NLL and MAE in predictions on a Weather time series benchmark. Their experiments also outperform a GP (with RBF kernel) on a multidimensional input/output task of simultaneous temperature, rainfall and wind speed regression task. Reviewers found the problem statement is novel and interesting, and the results are ample and promising. The paper can be improved by making experimental part clearer, and add analysis of runtime